# Emerging Targeted Therapies for HER2-Positive Breast Cancer

**DOI:** 10.3390/cancers15071987

**Published:** 2023-03-26

**Authors:** María Florencia Mercogliano, Sofía Bruni, Florencia Luciana Mauro, Roxana Schillaci

**Affiliations:** Instituto de Biología y Medicina Experimental (IByME-CONICET), Buenos Aires C1428ADN, Argentina

**Keywords:** HER2-positive breast cancer, monoclonal antibodies, tyrosine kinase inhibitors, immunotherapy, CAR-T cells, CAR-NK cells, CAR-M cells, cancer vaccines

## Abstract

**Simple Summary:**

HER2-positive breast cancer (BC), which accounts for ~20% of BC, is one of the more aggressive and has the worst overall survival rate among them. These patients are treated with trastuzumab, a monoclonal antibody targeting the HER2 molecule. Even though trastuzumab is an effective therapy, resistance events hamper its clinical benefit, making the development of new therapies a constantly growing area of interest. In this review, we will summarize the current therapies for HER2-positive BC and review the therapeutic approaches effective in preclinical models and clinical trials which could contribute to the therapeutic landscape. We will investigate the development of novel HER2-therapies such as antibodies, inhibitors, and bispecific antibodies, which is a burgeoning field in oncology. Furthermore, we will summarize the most recent developments in CAR-T, CAR-NK, and CAR-M therapies for the treatment of HER2-positive tumors, as well as a brief review of cancer vaccines.

**Abstract:**

Breast cancer is the most common cancer in women and the leading cause of death. HER2 overexpression is found in approximately 20% of breast cancers and is associated with a poor prognosis and a shorter overall survival. Tratuzumab, a monoclonal antibody directed against the HER2 receptor, is the standard of care treatment. However, a third of the patients do not respond to therapy. Given the high rate of resistance, other HER2-targeted strategies have been developed, including monoclonal antibodies such as pertuzumab and margetuximab, trastuzumab-based antibody drug conjugates such as trastuzumab-emtansine (T-DM1) and trastuzumab-deruxtecan (T-DXd), and tyrosine kinase inhibitors like lapatinib and tucatinib, among others. Moreover, T-DXd has proven to be of use in the HER2-low subtype, which suggests that other HER2-targeted therapies could be successful in this recently defined new breast cancer subclassification. When patients progress to multiple strategies, there are several HER2-targeted therapies available; however, treatment options are limited, and the potential combination with other drugs, immune checkpoint inhibitors, CAR-T cells, CAR-NK, CAR-M, and vaccines is an interesting and appealing field that is still in development. In this review, we will discuss the highlights and pitfalls of the different HER2-targeted therapies and potential combinations to overcome metastatic disease and resistance to therapy.

## 1. Introduction

Breast cancer is the most frequent tumor and the leading cause of death in women with cancer [1]. The HER2-positive (HER2+) subtype is defined by HER2 overexpression/amplification and accounts for ~20% of breast cancers [2], 25% of ovarian cancers [3], and 18% of gastric cancers [4]. HER2+ breast cancer, together with the triple-negative breast cancer (TNBC) subtype, have the worst 4-year survival rate and exhibit a more aggressive phenotype [5]. HER2 is an orphan membrane receptor, meaning it has no known ligand [6,7], of the HER family, which has three other members: HER1/EGFR, HER3, and HER4 [7]. When their specific ligands are present, the different family members can homo- or heterodimerize, resulting in activation of the PI3K/Akt, ERK (p42/p44 MAPK), and NF-B pathways [8,9], which result in tumor proliferation, migration, and survival [10,11].

The HER2+ breast cancer subtype is defined by assessment of HER2 expression by immunohistochemistry (score 3+ or 2+ with positive ISH), and the standard of care treatment is trastuzumab, a monoclonal antibody directed to the HER2 molecule [12], which is administered based on the results of HER2 expression. In the case of HER2+ breast cancer with positive hormone receptors (luminal B), endocrine therapy is incorporated into the trastuzumab and chemotherapy administration [13,14]. There are several prognostic biomarkers for HER2+ breast cancer, but no predictive biomarkers to determine which HER2-targeted therapy will be more effective for the patient. Figure 1 shows the first, second, and third lines of the current standard of care treatments.

HER2 expression measured by immunohistochemistry with a 2+ score without a positive ISH or 1+ score was considered HER2-negative until recently. Currently, a new HER2 entity is emerging: the HER2-low subtype, which is defined by 1+ or 2+ immunohistochemistry scoring with negative ISH. Despite the fact that trastuzumab has been shown to improve progression-free survival (PFS) and overall survival (OS) in HER2+ patients [12], resistance events limit its clinical benefit: 27–42% of patients progress to adjuvant or neoadjuvant trastuzumab, respectively, due to de novo or acquired resistance [15,16]. Given these clinical outcomes, significant efforts have been made to develop additional HER2-targeted therapies, such as monoclonal antibodies aimed at other HER2 epitopes, antibody-drug conjugates, bispecific antibodies, and tyrosine kinase inhibitors (TKIs), among others. Furthermore, these treatments are being tested in conjunction with cell therapies or drugs that target other molecules in order to achieve a synergistic effect, such as immune checkpoint inhibitors (ICIs), cyclin-dependent kinase (CDK) inhibitors, and PARP inhibitors, among others. In this review, we will focus on the standard-of-care treatments for HER2+ breast cancer patients, the current HER2-targeted therapies being tested in the clinical setting or those that have shown promising results in preclinical models, and potential combinations with other treatments, such as immunotherapies, to achieve cancer remission in HER2+ and HER2-low breast cancer subtypes. We will also discuss other treatment options, including adoptive cell therapies (CAR-T, CAR-M, CAR-NK, etc.) and cancer vaccines.

## 2. HER2-Targeted Therapies

### 2.1. Antibodies

#### 2.1.1. Monoclonal Antibodies

Trastuzumab has improved the treatment of HER2+ breast cancer patients since its introduction 25 years ago, and it has become the standard of care treatment due to its beneficial results. Trastuzumab is a monoclonal antibody that binds to the HER2 receptor’s IV domain, inhibiting downstream signaling and inducing antibody-dependent cell-mediated cytotoxicity (ADCC) [17,18] and antibody-dependent cell-phagocytosis (ADCP) [19]. Currently, trastuzumab is administered in the adjuvant setting in combination with chemotherapy or radiotherapy for one year. In the neoadjuvant setting and after the KRISTINE protocol, when tumors are bigger than 2 cm, trastuzumab is administered in combination with pertuzumab, a monoclonal antibody that targets the II domain of the HER2 molecule and inhibits its dimerization [20]. Pertuzumab, like trastuzumab, can interact with the Fc receptors of NK cells to unleash ADCC [21]. It has been demonstrated that first-line treatment with trastuzumab in combination with pertuzumab and taxanes produces comparable results regardless of the taxane used, implying that the latter can be substituted in the event of adverse effects [22,23,24,25].

Given the importance of trastuzumab for HER2+ breast cancer patients, several biosimilars have been introduced in the clinical setting. A biosimilar is a biopharmaceutical product that has similar active properties to a drug previously licensed, including structural and functional properties, and of course has to have comparable clinical efficacy [26,27]. The use of biosimilars creates not only different treatment regimens but also decreases the financial toxicity associated with the treatment, which is particularly attractive in the case of trastuzumab since it can be given at least for an entire year. Currently, there are five trastuzumab biosimilars, which have been extensively reviewed by Triantafyllidi et al. [28].

Due to the high rate of resistance to first-line treatment in HER2+ breast cancer patients, new HER2-targeted monoclonal antibodies are constantly being developed. Some of these novel monoclonal antibodies bind to the same epitopes as antibodies approved for clinical use, while others bind to different epitopes and can be modified to exert more potent ADCC. In the case of previously treated metastatic HER2+ breast cancer patients, the FDA approved the use of margetuximab in combination with chemotherapy [29]. Margetuximab binds the same epitope on the HER2 molecule as trastuzumab and with similar affinity, but the former has increased binding capacity to CD16A and reduced capacity to bind to CD32B, which consequently improves ADCC [30]. In the phase 3 clinical trial, margetuximab with chemotherapy exhibited better PFS when compared to trastuzumab plus chemotherapy in HER2+ advanced breast cancer [31]. However, the final analysis of the results showed no advantage over trastuzumab administration in OS survival [32]. It remains to be evaluated if margetuximab could be useful in patients with allelic variations in CD16A. In this sense, the MARGOT clinical trial (NCT04425018) is currently recruiting patients with HER2+ breast cancer who have low affinity CD16A alleles. Enhancement of ADCC in combination with HER2-targeted therapy is another strategy for developing new combined therapies for breast cancer treatment, and various strategies have previously been reviewed [33].

1E11 is a humanized monoclonal antibody that recognizes a different epitope on the IV domain on the HER2 molecule than trastuzumab that was tested in gastric cancer and presented synergistic effects in combination with trastuzumab [34], but it has not been tested in breast cancer patients. MM-302 consists of a HER2-targeted liposome with 45 anti-HER2 antibodies that contains 20,000 molecules of doxorubicin [35]. The combination of MM-302 and trastuzumab exhibited synergistic effects on breast and gastric cancer in xenograft models [35], but did not show additional benefits for patients in the HERMIONE clinical trial (NCT02213744).

Even though trastuzumab has been proven to be efficient in the treatment of HER2+ breast cancer, as it was mentioned before, resistance events impair its therapeutic effect [15,16]. Several trastuzumab resistance mechanisms have been described in the literature [36,37]. In particular, one of the described mechanisms of resistance is the expression of the membrane glycoprotein mucin 4 (MUC4), which masks the trastuzumab-binding epitope on the HER2 molecule [38]. In this regard, our group has demonstrated that tumor necrosis factor ɑ (TNFɑ) induces MUC4 expression in HER2+ breast cancer and converts trastuzumab-sensitive cell lines and tumors into resistant ones in vitro and in vivo [39,40]. Furthermore, we have demonstrated that INB03, a dominant negative protein that blocks soluble TNFɑ [41], sensitizes trastuzumab-resistant cell lines and tumors to the monoclonal antibody, resulting in an antitumor innate immune response [42]. TNFɑ-induced MUC4 expression, we believe, should be investigated as a potential resistance mechanism for trastuzumab-based and trastuzumab-like monoclonal antibodies. 

#### 2.1.2. Bispecific Antibodies

Bispecific antibodies (BsAb) are antibodies that can bind two different antigens on the same or a different molecule at the same time and are classified into two groups: the ones that have two Fabs and a Fc region (trifunctional antibodies) and the ones that do not have the latter. BsAbs exert their function through the Fab and Fc regions: ADCP, ADCC, complement-dependent cytotoxicity (CDC), inhibition of signaling pathways through interaction with membrane receptors, and induction of apoptosis. The current state-of-the-art on BsAbs has been extensively reviewed [43]. Since several BsAbs have been developed and tested throughout the years, in this section we will summarize those that target the HER2 molecule and have promising results in preclinical or clinical trials (Table 1).

##### HER2/HER2 BsAb

MBS301. This BsAb was engineered from trastuzumab and pertuzumab, and in vivo experiments showed that MBS301 had a more effective antitumor effect and removal of fucose enhanced ADCC than the monoclonal antibodies alone [44]. A phase 1 clinical trial is being conducted at present in malignant HER2-expressing solid tumors (NCT03842085).

Zanidatamab (ZW25). This is a pertuzumab and trastuzumab-based BsAb since it binds both of those domains in the HER2 molecule [45]. ZW25 is more effective than trastuzumab in binding to the tumor cells, inhibiting cell growth, receptor internalization, and downregulating HER2 expression in various HER2-expressing tumors, including HER2+ and HER2-low breast cancer [46,47]. The clinical trial of ZW25 (NCT02892123) has already analyzed the maximum tolerated and optimal dose in patients with HER2+ solid tumors [48], and its combination with chemotherapy showed promising results [49,50]. The pharmacokinetics of ZW25 have recently been investigated [43,51]. In this sense, there are currently several trials evaluating ZW25 in different scenarios: ZW25 in early breast cancer (NCT05035836) and for advanced breast cancer in combination with palbociclib and fulvestrant (NCT04224272), ZW25 in combination with anti-PD1 (Programmed cell death protein 1 PD-1) in gastric cancers (NCT05270889 and NCT05152147), ZW25 plus anti-CD47 in HER2+ solid tumors including HER2-overexpressing breast cancers as well as HER2-low breast cancers (NCT05027139), ZW25 in biliary tract cancer (NCT04578444, NCT04466891, and NCT05615818), among others.

KN026. This is another BsAb based on trastuzumab and pertuzumab that inhibits proliferation of HER2-expressing cells and tumors and is effective in trastuzumab-/pertuzumab-resistant models as well [52]. The phase 1 clinical trial showed good tolerability, and KN026 exhibited similar efficacy to the combination of trastuzumab and pertuzumab in HER2+ metastatic breast cancer patients [53]. This report also underscores the amplification of CDK12 as a potential biomarker of response to KN026. There are currently 9 trials for HER2+ solid tumors. Regarding breast cancer, KN026 is being studied in the neoadjuvant setting in combination with chemotherapy (NCT04881929), in combination with KN046 (a new BsAb targeting PD-1, CTLA-4, NCT04521179, and NCT04040699), which demonstrated to be safe and had antitumor activity in preliminary results presented recently [54], and in combination with palbociclib and fulvestrant (NCT04778982).

##### HER2/HER3 BsAb

Zenocutuzumab (MCLA-128). This BsAb not only blocks HER2/HER3 signaling [55,56] but also promotes tumor elimination through ADCC [57], even in HER2-low cell lines [58] or cell lines with low affinity for CD16 [59]. The phase 1/2 of zenocutuzumab (NCT02912949) indicated that it was well tolerated, and a promising antitumor response was obtained in heavily pretreated breast cancer patients [60]. A phase 1 study is currently studying dose escalation, tolerability, safety, and antitumor activity of zenocutuzumab in NRG1 fusion-positive cancers (NCT02912949). Another phase 2 trial is testing Zenocutuzumab in combination with chemotherapy, trastuzumab, and endocrine therapy for HER2-low and estrogen receptor-positive breast cancers, respectively (NCT03321981). Zenocutuzumab was also tested in combination with trastuzumab and vinorelbine in metastatic breast cancer that progressed to trastuzumab-emtansine (T-DM1). This study showed that the combination was well tolerated, with adverse events mostly related to the chemotherapy, but more importantly, the treatment was effective in metastatic HER2+ breast cancer patients [61].

MM-111. A BsAb that binds to HER2 and HER3 and inhibits the activation of HER3 and PI3-K signaling pathways [62]. Its combination with trastuzumab or lapatinib had antitumor effects in preclinical models [62] and in a phase 1 clinical trial that showed a 55% clinical benefit rate in HER2+ solid tumors [63]. Subsequent trials have been completed, but results are currently not available (NCT01097460 and NCT00911898).

**Table 1 cancers-15-01987-t001:** Bispecific antibodies for HER2+ breast cancer in clinical trials.

Drug	Clinical Trial Identifier	In Combination with	Population	Reference
Trastuzumab/Pertuzumab				
MBS301	NCT03842085		Malignant HER2-expressing solid tumors	[44]
Zanidatamab (ZW25)	NCT02892123	Chemotherapy	HER2-expressing solid tumors	[49,50]
	NCT05035836		Early HER2+ breast cancer	
	NCT04224272	Palbociclib and fulvestrant	Advanced HER2+ breast cancer	
	NCT05027139	Anti-CD47	Solid HER2+ tumors including the HER2-low breast cancer	
KN026	NCT04881929	Chemotherapy	HER2+ breast cancer	
	NCT04521179 NCT04040699	KN046 (bispecific antibody against PD-1 and CTLA-4)	Locally advanced HER2+ solid tumors and HER2+ solid tumor	[54]
	NCT04778982	Palbociclib and fulvestrant	Advanced breast cancer	
**HER2/HER3**				
Zenocutuzumab (MCLA-128)	NCT03321981	Trastuzumab and chemotherapy or trastuzumab and vinorelbine	HER2-low breast cancer and metastatic HER2+ breast cancer that progressed to T-DM1 treatment	[61]
MM-111	NCT01097460	Trastuzumab	Advanced HER2 amplified and heregulin-positive breast cancer	
	NCT00911898		Advanced, refractory HER2 A\amplified and heregulin-positive cancers	

##### HER2 and CD3 BsAb

Ertumaxomab. BsAb that targets HER2 (identifies a different epitope than trastuzumab or pertuzumab) and CD3, and preferentially binds to Fcγ receptors [64], resulting in a trifunctional antibody that recognizes T cells, stromal cells, and tumor cells, and has an antitumor effect in HER2-high or HER2-low breast tumors [65]. This BsAb has been evaluated in phase 1 clinical trials and exhibited a moderate antitumor effect in heavily pretreated breast cancer that could not compete with the standard of care and presented severe side effects [64,66].

p95HER2. This truncated HER2 is only expressed in tumors, and a BsAb was generated to target CD3 as well. This BsAb exhibited a strong antitumor effect on HER2+ primary breast cancer and brain lesions in vitro and in vivo [67].

GBR1302. This is a BsAb that binds to the invariant CD3 chain of the TCR and directs T cells to HER2+ breast cancer cells. The phase 1 clinical trial initially showed an increase in cytokine production and positive regulation of T cells, but the trial is currently halted and will not resume [68].

##### HER2 and CD16 BsAb

HER2 and CD16 BsAb. HER2(Per)-S-Fab. This BsAb consists of a pertuzumab Fab and an anti-CD16/FcγRIIIA antibody with exceptional in vitro and in vivo cytotoxicity [69]. This report also indicated that this BsAb could be effective in HER2-low breast cancer and could overcome trastuzumab resistance. Turini et al. reported that the HER2bsFab BsAb exhibited moderate affinity for HER2 and high affinity for FcγRIII, which resulted in effective ADCC in HER2 overexpressing cell lines but also worked in HER2-low and even in trastuzumab-resistant cell lines [70]. Li et al. also produced a BsAb using the Fab from trastuzumab and a single domain of anti-CD16 VHH, which showed good results in vivo and in vitro [71]. Interestingly, a tribody monoclonal antibody ((HER2)2XCD16) was designed to obtain an anti-HER2 single chain variable fragment fused to IFN-γ that causes IFN-γ receptor-dependent apoptosis even in cells and tumors resistant to HER2-targeting. The tribody exhibited a greater effect on cell lysis generated by γδ T cells and NK cells not only in HER2-expressing models but also in breast cancer [72]. Moreover, the tribody altered the tumor microenvironment (TME) and converted it to an antitumor TME [73].

Lastly, the main adverse events for bispecific antibodies consist of cytokine release syndrome (CRS) of grades 3 and 4 [74,75], which has been suggested to be avoided by retention of the Fc portion of the antibodies. The development of BsAb is a rapidly growing field, which is reflected in the plethora of antibodies being generated and tested in clinical and preclinical models. In the next few years, the ongoing clinical trials will be completed and could change the management of HER2+ breast cancer patients in the clinic.

#### 2.1.3. Antibody-Drug Conjugates (ADCs)

This therapeutic approach takes advantage of the specific targeting of monoclonal antibodies and combines it with drugs with potent cytotoxic effects, achieving targeted drug delivery [76]. The ADCs as a whole have a synergistic effect when compared to their parts alone; this is mainly due to the bystander killing effect, in which the drug payload can exert its effect not only on the target cells but also in the TME [77]. In addition, the ADCs have demonstrated that they are effective even when the target protein is expressed in small amounts. In HER2+ breast cancer, T-DM1 made its debut to revolutionize the field [78], and ADCs are being developed in hopes of better treatment options. In this sense, ADCs have shown promising results against brain metastases [79] and in HER2-low breast cancer [80]. There are currently 14 FDA-approved ADCs for various cancers, with the target molecule, conjugated drugs, and/or the linkers varying. Adverse events of ADCs account for 10–15% [81,82], and the most common are fatigue, neuropathies, leukopenia, thrombocytopenia, pneumonitis, interstitial lung disease, and nausea [83,84,85]. In this section, we will summarize the main trastuzumab-based ADCs used in the clinical setting shown in Table 2 and those that exhibit promising results in preclinical studies for HER2+ breast cancer.

T-DM1. This was the first FDA-approved ADC for the adjuvant and neoadjuvant treatment of HER2+ breast cancer patients who had progressed to trastuzumab and a taxane [81,86,87,88]. This ADC consists of trastuzumab loaded with an average of 3.6 molecules of emtansine, a microtubule inhibitor [89]. Currently, T-DM1 is used as a second-line treatment or as a first-line option when patients cannot obtain or are not suitable for the standard of care (trastuzumab plus pertuzumab with chemotherapy or radiotherapy) [90]. Regarding brain metastasis, the EMILIA [91] and KAMILLA [92] trials showed clinical benefit for patients treated with T-DM1. Even though T-DM1 has had good results, resistance events hamper its efficiency almost always in first responders [93], and resistance mechanisms have already been reviewed [36,94]. Particularly for HER2-low breast cancer, T-DM1 showed limited efficacy, but there are no prospective clinical trials [95].

Trastuzumab-deruxtecan (T-DXd). This ADC results from the combination of trastuzumab with eight molecules of a topoisomerase I inhibitor (deruxtecan), which induces apoptosis and double-strand DNA breaks [96]. T-DXd outperformed T-DM1 in HER2+ and HER2-low breast cancer preclinical models due to payload permeability and the bystander killing effect [97]. The phase 1 clinical trial of T-DXd showed an acceptable safety profile and a potent antitumor effect in heavily pretreated HER2 breast cancer patients [98,99]. Moreover, T-DXd was effective in HER2+ breast cancer patients previously treated with T-DM1 [100,101]. At the end of 2019, T-DXd was granted an accelerated FDA approval for advanced, metastatic, or unresectable HER2+ breast cancer based on the DESTINY-Breast01 [82] and the DESTIINY-Breast03 (NCT03529110) [102,103]. Given the promising results of T-DXd in the clinical setting, several other trials were launched to test this ADC to treat other HER2-expressing cancers, such as gastric (NCT05034887 and NCT04989816), lung (NCT05246514 and NCT05048797), colorectal (NCT04744831), bladder, urothelial, or endometrial (NCT04639219 and NCT04482309). Furthermore, the use of T-DXd in combination with other therapies is also being explored, such as in combination with pertuzumab in metastatic breast cancer (NCT04784715), durvalumab (anti-PD-L1 and NCT04538742), tucatinib (TKI targeting HER2, NCT04538742, and NCT04539938) in metastatic HER2+ breast cancer, and even in HER2-low breast cancer in combination with several distinct drugs (NCT04556773) [104]. The preliminary results of T-DXd in HER2-low breast cancer showed such great results and tolerability (NCT02564900) [105,106] that T-DXd is now recommended in the ASCO guidelines. T-DXd’s success has resulted in several clinical trials testing its combination with ICIs, endocrine therapy, and chemotherapy, among other things, in HER2-low breast cancer (NCT04556773).

Trastuzumab-duocarmycine/SYD985. This is the most developed ADC after T-DXd and consists of trastuzumab conjugated to duocarmycine, an irreversible DNA alkylating agent for dividing and non-dividing cells with a potent bystander killing effect [107]. Early clinical studies yielded favorable results [84,85,108], resulting in the TULIP phase 3 trial (NCT03262935), where preliminary data analysis shows that SYD985 improves PFS compared to physician-assisted chemotherapy [109]. In the I-SPY clinical trial (NCT01042379), SYD985 is being tested in combination with several drugs plus chemotherapy in the neoadjuvant setting, with paclitaxel (NCT04602117) or with PARP inhibitors (NCT04235101) in HER2+ or HER2-low breast cancer. With regard to the latter, thanks to the bystander effect, SYD985 is very effective in HER2-low breast cancer [83,84,85].

ARX788. ARX788 is a novel next-generation ADC conjugated with a non-cleavable linker to AS269 (Amberstatin 269, a synthetic dolastatin), a hydrophilic payload with limited permeability that results in a decreased bystander effect [110,111]. In preclinical models, ARX788 exhibited antitumor activity in HER2+ and HER2-low breast cancers and even in T-DM1-resistant models [110,112]. The results from the phase 1 trial in patients with metastatic breast cancer showed low toxicity and promising effectiveness [113]. Currently, there are nine clinical trials registered and evaluating ARX788 in HER2+ breast (NCT01042379) and metastatic breast cancer (NCT04829604 and NCT02512237), HER2-mutated tumors (NCT05041972), HER2-low breast cancer (NCT05018676), breast cancer patients with brain metastasis (NCT05018702), and HER2+ solid tumors (NCT03255070).

Disitamab vedotin. This novel ADC, also called RC48, is composed of hertuzumab linked to vedotin, which consists of the linker and 4 molecules of the cytotoxic agent monomethyl auristatin E (MMAE). Vedotin has already been tested and approved by the FDA in 2011 for the treatment of lymphoma [114,115], and it has been approved in China for gastric and metastatic breast cancer [116,117]. RC48 exhibited a good performance in HER2+ gastric, breast, and urothelial preclinical models [116]. RC48 offers a more potent bystander effect and antitumor activity when compared to T-DM1, even in trastuzumab-resistant models [118]. A phase 1 clinical trial in patients with advanced or metastatic HER2+ tumors showed that RC48 monotherapy had promising results and was well tolerated (NCT02881190), particularly in patients with HER2-low gastric cancer [119,120] and advanced or metastatic breast cancer [121]. Moreover, a phase 2 study (NCT03556345) in patients with advanced or metastatic gastric cancer treated with two or more lines of chemotherapy showed a 24.8% objective response rate [122], which led to the conditional approval of RC48 administration for gastric cancer patients. There are 48 trials studying RC48 in different cancer types; in breast cancer, there are currently six studies exploring de-escalation, combinatorial therapies, effects on liver metastasis, and HER2-low-expressing breast cancers (NCT05134519, NCT04400695, NCT05331326, NCT03052634, NCT05726175, and NCT03500380).

A166. A166 is a trastuzumab-based ADC linked to duostatin-5, an anti-microtubule agent, which in a phase 1/2 clinical trial demonstrated good tolerability and a promising antitumor effect in heavily pretreated breast cancer patients (NCT03602079) [123,124,125].

MRG002. MRG002 consists of a humanized anti-HER2 with a linker that carries ~3.8 MMAE [126] that showed tolerable toxicity and an antitumor effect in HER2+ breast cancer PDX and in mouse xenografts that was more efficient than trastuzumab and T-DM1, since it exhibited results even in T-DM1-resistant models [126]. Currently, there are three clinical trials in HER2+ or HER2-low breast cancer patients exploring the MRG002 effect in advanced (NCT05263869) and metastatic tumors (NCT04924699 and NCT04742153).

Zanidatamab zovodotin (ZW49). ZW49 is an ADC based on the above-mentioned ZW25, a bispecific antibody composed of trastuzumab and pertuzumab binding epitopes conjugated to auristatin that has higher internalization than the monospecific trastuzumab ADC [127]. At a tolerable toxicity, ZW49 exhibited antitumor activity in brain metastases as well as in HER2-overexpressing and HER2-low PDX models [127]. ZW49 is currently being tested in a phase 1 clinical trial in patients with metastatic HER2+ tumors (NCT03821233).

BDC-1001. This ADC consists of a trastuzumab biosimilar and a TLR7/8 agonist, which stimulates antigen-presenting cells (APCs). BDC-1001 induced a robust antitumor immune response in preclinical models, prompting the first-in-human clinical trial to test this molecule in advanced HER2+ tumors in combination with nivolumab (NCT04278144). Preliminary analysis of this trial indicated acceptable tolerability and safety [128].

ALT-P7. This is a novel ADC composed of the trastuzumab biobetter HM2 conjugated to MMAE [129]. A phase 1 clinical trial of ALT-P7 in advanced HER2+ breast cancer patients (NCT03281824) is being conducted, and preliminary results show good tolerability at the lowest dose, warranting the development of a phase 2 [130].

XMT-1522. This ADC is constituted by HT-19, an antibody that binds to the IV domain of HER2, distinct from the trastuzumab epitope, conjugated with 12 molecules of an auristatin derivative (AF-HPA) [131]. XMT-1522 has been shown to be effective in mouse models bearing HER2+ breast and gastric tumors and even on T-DM1-resistant models [132]. There is currently only one phase 1 clinical trial testing this molecule in advanced HER2+ breast cancer patients (NCT02952729), with preliminary data indicating good tolerability and antitumor activity [133].

PF-06804103. This ADC is composed of a trastuzumab-derived antibody and a potent auristatin derivative [134,135], which has been shown to have antitumor activity in HER2+ and HER2-low breast, lung, and gastric cancer [135,136]. A phase 1 clinical trial in heavily pretreated patients with HER2+ tumors (NCT03284723) showed manageable toxicity and a promising antitumor effect [136].

Alta-ADC. This is an engineered pertuzumab-based ADC that has only been tested in vitro and in vivo. ALTA-ADC has a lower affinity for HER2 at low pH [137]. This is useful because the internalized ADC dissociates from HER2 in the endosome, where the payload can be released into the lysosomes and the antibody can bind to an unbound target. Due to this, Alta-ADC is effective at lower doses, which may be useful in avoiding toxicity events. When compared to T-DM1, ALTA-ADC demonstrated a significant antitumor effect in preclinical models. Targeted thorium-227 conjugates (TTCs)/BAY2701439. TTCs are composed of 𝛼-emitting radionuclides (thorium-227) that induce double strand breaks, resulting in cell elimination; they also stimulate the release of danger-associated molecular patterns that are recognized by the immune system, resulting in immunogenic death [138]. The TTCs can be directed to the tumor using a targeted therapy approach [139]. In preclinical models, HER2-TTC exhibited a selective antitumor effect in several HER2+ lung, gastric, bladder, and breast cancers. In the latter, mouse models encompassed tumors with different HER2 expression, including the HER2-low entity and T-DM1-resistant models. BAY2701439 consists of a trastuzumab-derived antibody conjugated to TTCs and is currently being tested in patients with HER2+ breast, gastric, or gastroesophageal tumors (NCT04147819).

**Table 2 cancers-15-01987-t002:** Current clinical trials of ADCs in HER2+ breast cancer.

Drug	Payload	Drug-to-Antibody Ratio	Clinical Trial Identifyer	In Combination with	Population	Reference
T-DXd	Deruxtecan (topoisomerase I inhibitor)	~8	NCT04784715	Pertuzumab	HER2+ metastatic breast cancer	
			NCT04538742	Durvalumab (anti-PD-L1,)	HER2+ metastatic breast cancer	
			NCT04538742, NCT04539938	Tucatinib	HER2+ breast cancer or HER2+ metastatic breast cancer	
			NCT04556773	Durvalumab, paclitaxel, capivasertib, anastrozole, fulvestrant, or capecitabine	HER2-low advanced or metastatic breast cancer	[104]
Trastuzumab-duocarmycine (SYD985)	Duocarmycine (DNA alkylating agent)	2.8	NCT03262935		HER2+ locally advanced or metastatic breast cancer	[109]
			NCT01042379 (I-SPY)	Chemotherapy	Breast cancer	
			NCT04602117 (ISPY-P1.01)	Paclitaxel	Metastatic cancer	
			NCT04235101	Niraparib (PARP inhibitor)	Solid tumors	
ARX788	Amberstatin 269 (microtubule inhibitor)	2	NCT01042379		HER2+ breast cancer	
			NCT04829604, NCT02512237		HER2+ metastatic breast cancer	
			NCT05041972		HER2-mutated or HER2-amplified tumors	
			NCT05018676		HER2-low breast cancer	
			NCT05018702		Breast cancer patients with brain metastasis	
			NCT03255070		HER2+ solid tumors	
Disitamab vedotin (RC48)	Monomethyl auristatin E (microtubule inhibitor)	4	NCT02881190		Advanced or metastatic HER2+ tumors	[121]
			NCT05134519		HER2+ breast cancer	
			NCT04400695		Locally advanced or metastatic HER2-low breast cancer	
			NCT05331326		HER2-expression metastatic breast cancer with abnormal activation of PAM pathway	
			NCT03052634		Advanced breast cancer	
			NCT05726175	Penpulimab (AK105)	HER2-low breast cancer	
			NCT03500380		HER2+ metastatic breast cancer with or without liver metastases	
A166	Duo-5 (microtubule inhibitor)	2.8	NCT03602079		Relapsed/refractory cancers wxpressing HER2 antigen or amplified HER2 gene	[123,124,125]
MRG002	Monomethyl auristatin E (microtubule inhibitor)	~3.8	NCT05263869		HER2+ advanced breast cancer	
			NCT04924699		HER2+ metastatic tumors	
			NCT04742153		HER2-low locally advanced metastatic breast cancer	
Zanidatamab zovodotin (ZW49)	Auristatin based (microtubule inhibitor)	2	NCT03821233		Metastatic HER2+ tumors	
BDC-1001	TLR7/8 agonist	Not reported	NCT04278144	Nivolumab	Advanced HER2-expressing solid tumors	[128]
ALT-P7	Monomethyl auristatin E (microtubule inhibitor)	2	NCT03281824		HER2+ breast cancer	[130]
XMT-1522	Auristatin derivative (AF-HPA)	12	NCT02952729		Advanced HER2+ breast cancer patients	[133]
PF-06804103	Derivative of auristatin	4	NCT03284723		HER2+ breast cancer	[136]
Targeted thorium-227 conjugates (TTCs)/BAY2701439	Thorium-227 (cytotoxic alpha radiation)	Not reported	NCT04147819		Advanced HER2-expressing cancer	

### 2.2. TKIs

TKIs are a family of small molecules that inhibit protein tyrosine kinases, which are responsible for signal transduction to regulate cellular, physiological, and biochemical processes. TKIs compete with ATP in tyrosine kinase receptors’ ATP binding domain, inhibiting downstream signaling [140]. In this sense, TKIs can inhibit cell proliferation, migration, and invasion and induce apoptosis. Due to their size, TKIs can cross the blood-brain barrier more easily than other HER2-targeted therapies such as monoclonal antibodies or ADCs [141]. Several TKIs that target the HER2 family are currently approved for the treatment of cancer in combination with other therapies, and this area is still being researched due to resistance events that impair their effectiveness and off-target toxicity. In this section, we will describe the TKIs currently used in the clinic and novel inhibitors that have been successful in preclinical or clinical trials and could offer therapeutic alternatives.

Lapatinib. This was the first TKI approved for breast cancer with high or low HER2 expression [142]. Is a reversible small molecule inhibitor of EGFR/HER1 and HER2 that blocks the phosphorylation of the receptors, which inhibits the activation of the MAPK and PI3-K pathways and consequently inhibits cell proliferation [143]. The NeoALTTO clinical trial demonstrated that the combination of trastuzumab and lapatinib in the neoadjuvant setting exhibited a higher pathological complete response (pCR) in patients with HER2+ breast cancer, but overall survival and disease-free survival presented no significant differences [144,145,146]. It is noteworthy to mention that the adverse effects for lapatinib and TKIs in general imply diarrhea, rash, infections, and hepatic toxicity [147,148], and in the NeoALTTO trial, 65% and 31%, respectively, of the enrolled patients in the neoadjuvant and adjuvant settings had to discontinue the treatment due to adverse events [145]. Lapatinib showed penetration of the blood-brain barrier and exhibited a reduction in brain metastases, as it was reported in the EGF105084 [149], LANDSCAPE [150], and CEREBEL [151] clinical trials. In this regard, Khan et al. reported intracranial activity of lapatinib and increased survival for HER2+ breast cancer patients with brain metastases in a meta-analysis [152]. Currently, lapatinib is FDA-approved for breast cancer treatment in combination with letrozole or capecitabine in hormone receptor positive and HER2+ breast cancer, or with capecitabine or with trastuzumab for receptor positive and HER2+ postmenopausal patients, in which case an increase in PFS was observed but no benefit in overall survival [153]. Lapatinib is not currently approved for its use in the neoadjuvant setting.

Neratinib. This is an irreversible pan-HER inhibitor that not only has a greater effect than lapatinib but also can increase trastuzumab-mediated ADCC [154,155,156,157]. Preclinical studies showed that neratinib induces cell cycle arrest and proliferation inhibition in HER2-expressing cells [158]. In the exteNET clinical trial, neratinib showed a 73% rate of response in patients with HER2+ metastatic breast cancer previously treated with adjuvant or neoadjuvant trastuzumab, but on the other hand, it exhibited high toxicity, which led to dose reduction and treatment of the adverse events, the most commonly reported of which is diarrhea [159,160,161,162]. When tested in patients with early HER2+ breast cancer, the results were as promising as for the advanced cases, since neratinib showed an improvement in disease-free survival [163]. Neratinib in combination with lapatinib showed better tolerability and effectiveness than lapatinib alone [164]. Neratinib is currently being tested in combination with other therapies such as T-DM1 (NCT02236000) [165] and fulvestrant (NCT03289039) [166,167]. Regarding brain metastases, neratinib showed a reduced incidence of central nervous system events as presented in the NEfERT-T, NALA, and TBCRC 022 trials [168,169,170]. Given these clinical trial results, adjuvant neratinib was FDA-approved in 2017 for patients with HER2+ breast cancer who were previously treated with adjuvant trastuzumab for one year, and it is also administered in combination with capecitabine since 2020 for patients who received at least two previous anti-HER2 treatments [169,171].

Tucatinib. This is a selective HER2 TKI with reduced inhibition on EGFR that exhibited antitumor activity in breast and gastric tumors in preclinical models administered as a monotherapy [172] or in combination with trastuzumab in HER2+ breast cancer xenograft models [173]. These results led to the development of a phase 1 clinical trial (HER2CLIMB and NCT02614794) to test tucatinib in combination with trastuzumab and capecitabine, which showed great antitumor effect in metastatic breast cancer with the only adverse events being diarrhea, nausea, and palmo-plantar erythrodysesthesia [174]. Given the results of the phase 3 HER2CLIMB [175] trial, tucatinib was approved by the FDA in 2020 to treat patients with metastatic HER+ breast cancer. Tucatinib also showed improved efficacy in reducing brain metastasis [176,177]. Considering these results, tucatinib is the first TKI that was approved by the FDA for the treatment of brain metastases. Due to the promising results of tucatinib, there are several clinical trials exploring the effect of the combination of tucatinib with T-DM1 (NCT04457596, NCT03975647, NCT01983501, and NCT05323955), with T-DXd (NCT04539938 and NCT04538742), and with CDK4/6 inhibitors (NCT03054363) in HER2+ breast cancer patients.

Pyrotinib. This is an irreversible pan-HER TKI that can block cell cycle progression and inhibit tumor proliferation [178,179]. There are a plethora of ongoing clinical trials testing pyrotinib efficacy, and two studies that have already been completed (NCT01937689 and NCT02361112). In the NCT01937689 clinical trial, pyrotinib was administered alone to HER2+ metastatic breast cancer patients and showed a promising antitumor effect, with diarrhea and neutropenia being the main adverse events [178]. In the NCT02361112 clinical trial, pyrotinib was combined with capecitabine in HER2+ metastatic breast cancer patients, and the combination demonstrated good tolerability and antitumor activity [180,181,182]. Given these results, pyrotinib was approved in China in 2018 for the treatment of advanced or metastatic breast cancer [183]. Pyrotinib showed a better objective response rate and progression-free survival than lapatinib in a phase 2 clinical trial [181,183] and in patients with HER2+ metastatic breast cancer previously treated with trastuzumab plus chemotherapy [182]. Pyrotinib has also been shown to have an effect on brain metastases [182]. Even though pyrotinib has exhibited good results in the treatment of HER2+ breast cancer patients, no countries other than China have been exploring its application (Table 3).

Poziotinib. Poziotinib is an irreversible pan-HER TKI that has demonstrated antitumor activity in HER2+ cancer cell lines [184] and in patients in a phase 1 clinical trial [185]. Poziotinib upregulates HER2 expression and has a synergistic effect with T-DM1 [186]. In a phase 2 trial, patients with HER2+ metastatic breast cancer who had received at least two lines of HER2-targeted therapy were given poziotinib, which demonstrated promising antitumor activity in these heavily pretreated patients [187,188,189]. The main adverse events reported were diarrhea and stomatitis. The combination of poziotinib with T-DM1 is currently being tested (NCT03429101).

Epertinib (S-222611). This is a reversible pan-HER inhibitor that showed more potent effects than lapatinib in vitro and in vivo [190] and presented promising results in an HER2+ brain metastases cancer model [191]. Epertinib was tested in a phase 1 clinical study in patients with solid cancers, including HER2+ tumors, that showed good tolerability and safety as well as effective antitumor activity, especially in brain metastases [192,193,194].

DZD1516. This is a reversible and selective HER2 inhibitor that showed tumor regression in preclinical models of breast cancer and brain metastases [195]. A phase 1 clinical trial is being conducted to test the effect of DZD1516 in combination with trastuzumab and capecitabine or with T-DM1 (NCT04509596). Preliminary data shows that DZD1516 has full blood-brain barrier penetration and a good safety profile. Interestingly, diarrhea has not been reported as an adverse event in this study [196].

**Table 3 cancers-15-01987-t003:** Current clinical trials of selected TKIs in HER2+ breast cancer.

Drug	Description	In Combination with	Clinical Trial Identifyer	Population	Reference
Tucatinib	Selective and reversible HER2 inhibitor with minimal inhibition of EGFR/HER1	T-DM1	NCT04457596, NCT03975647, NCT01983501, NCT05323955	HER2+ breast cancer	
		T-DXd	NCT04539938, NCT04538742	HER2+ breast cancer	
Pyrotinib	Irreversible pan-HER inhibitor		NCT01937689	HER2+ metastatic breast cancer	[178]
		Capecitabine	NCT02361112	HER2+ metastatic breast cancer	[180]
Poziotinib	Irreversible pan-HER inhibitor	T-DM1	NCT03429101	HER2+ breast cancer	
Epertinib (S-222611)	Reversible pan-HER inhibitor		2013-003894-87	HER2+ tumors	[192,193,194]
DZD1516	Selective HER2 inhibitor	Trastuzumab and capecitabine or T-DM1	NCT04509596	Metastatic HER2+ breast cancer	[196]

## 3. HER2-Targeted Therapies in the Era of Immunotherapy

In spite of the efforts for developing new strategies against HER2, there are still a 20% of the patients with local disease who experience de novo or acquired resistance to the HER2-targeted therapies [15,197]. In particular, the in vivo mechanism of action of trastuzumab and trastuzumab-based therapies relies on the innate and adaptive immune response [18]. ADCC and ADCP, mainly performed by NK cells and macrophages, respectively, trigger an innate immune response that promotes antigen presentation and the subsequent adaptive immune response [19,198,199,200,201]. This evidence and preclinical data point out that using ICI enhances the trastuzumab antitumor effect, providing the rational basis for the combination of ICI with HER2-targeted therapies to overcome therapy resistance [202].

### 3.1. PD-1/PD-L1 Antibodies

The therapeutic possibility of reinvigorating the antitumor immune response through antibodies that inhibit effective T cell recognition of cancer cells has been a breakthrough in the oncology arena. Antibodies against PD-1, expressed in T cells, and PD-L1, its ligand present in tumor cells, interrupt the signals that promote T cell exhaustion. PD-L1 (Programmed Cell Death Ligand 1) is also expressed in APCs and can form a cis-heterodimer with the costimulatory molecule CD80 on the same cell, modulating the PD-1/PD-L1 interaction [203]. In particular, in TNBC, there are hundreds of clinical trials pursuing the administration of ICI, in particular those targeting PD-1 or PD-L1. Indeed, pembrolizumab has proved to be effective and has been approved by the FDA for the treatment of metastatic and high-risk early stage TNBC [204]. The scenario is different in HER2+ tumors, where ongoing clinical trials using immunotherapy in combination with HER2-targeted therapies are about 24 (Table 4) and few of them have reported results. Here, we briefly describe the published results of these trials. All the completed, recruiting, or active trials are listed in Table 4.

Pembrolizumab is a humanized IgG4k monoclonal antibody that binds to the PD-1 receptor. It impedes the interaction of PD-1 with its ligands, PD-L1 and PD-L2 [213]. The first clinical trial to address the benefit of combining the antitumor effect of trastuzumab with pembrolizumab in advanced trastuzumab-resistant HER2+ breast cancer was PANACEA. This was a single-arm, multicenter, phase 1b/2 trial that recruited 6 patients for phase 1 and 54 for phase 2. The combination of both antibodies showed to be safe and have clinical benefit in 6 out of 40 patients whose tumors were PD-L1-positive [205]. Pembrolizumab was also combined with T-DM1 in the NCT03032107 trial. This was a single-arm phase 1b study and included 20 metastatic breast cancers previously treated with trastuzumab and pertuzumab but not with T-DM1. The objective response rate was 20%, and PD-L1 expression had no correlation with treatment response [206].

Atezolizumab is a humanized IgG1κ immunoglobulin monoclonal antibody that binds to PD-L1, preventing its interaction with PD-1 and CD80 [214]. The KATE trial studied the addition of atezolizumab to T-DM1 in metastatic HER2+ breast cancer patients. This was a randomized, double-blind, placebo-controlled phase 2 trial that treated 133 patients with atezolizumab plus T-DM1 and 69 with T-DM1 plus placebo. This trial did not reach a significant advantage for atezolizumab with T-DM1 in progression-free survival considering all patients, but a better result was obtained by stratifying patients scored as PD-L1-positive. The objective response of the PD-L1-positive patients was 54% in the atezolizumab group vs. 33% in the placebo group. On the other hand, the PD-L1 negative patients exhibited a 39% objective response in the atelolizumab arm vs. 50% in the placebo arm. In this case, PD-L1 positivity is considered when 1% or more of tumor-infiltrating immune cells express PD-L1 [207]. The IMpassion050 phase 3 trial was performed to treat high-risk early breast cancer with atezolizumab (228 patients)/placebo (226 patients) with dose-dense chemotherapy and trastuzumab plus pertuzumab in the neoadjuvant setting. Unfortunately, regardless of PD-L1 expression in tumor-infiltrating immune cells, there was no difference in pCR rates between the atezolizumab and placebo groups [208].

Durvalumab is a fully human IgG1κ monoclonal antibody that targets PD-L1 [215]. Durvalumab plus trastuzumab was used to treat metastatic HER2+ patients in the phase 1 CCTG IND.229 clinical trial. The cohort of 14 patients was PD-L1 negative in their tumor cells. The results showed that the combination treatment was safe, but no objective response was obtained [210].

Avelumab is a fully human IgG1λ monoclonal antibody directed against PD-L1. The JAVELIN solid tumor trial is a phase 1b trial that recruited 168 metastatic breast cancer patients refractory to or progressing after standard of care therapy, of whom 26 were HER2+. The patients were treated with avelumab as monotherapy, rendering no objective response in all the HER2+ cases [211].

As it was mentioned before, another emerging strategy is the use of bispecific antibodies. There are a lot of efforts towards targeting HER2 and blocking the PD-1/PD-L1 interaction to simultaneously inhibit HER2 signaling and reinvigorate T cell function within the TME. One of these approaches is BsPD-L1xrErbB2. A Fc IgG2a fusion protein fused to the VH coding regions of rat Her2 and PD-L1.The BsAb exerted its effect on HER2+ mouse mammary carcinoma through CD8+ T cells and IFN-γ and achieved an increased CD8+ tumor infiltration [216]. SSGJ-705 consists of an anti-HER2/anti-PD-1 BsAb that links T cells and the tumor’s HER2-expressing cells and inhibits tumor proliferation. Through the inhibition of PD-1 signaling, it can restore immune function by activating T cells. A phase 1 clinical trial has been proposed in 2021, but recruitment has yet to begin (NCT05145179). The BsAb anti-HER2×PD1 was generated by fusion of the anti-PD-1 antibody (SSGJ-609A) and trastuzumab and showed promising antitumor activity in vitro and in vivo [217]. The anti-HER2×PD1 not only activated T cells and induced ADCC to eliminate HER2-expressing tumor cells but also linked tumor cells and T cells, forming an immunological synapse that achieved tumor killing that did not require antigen presentation [217]. Another BsAb is HER2/PD-L1. It consists of whole trastuzumab IgG1, which can trigger ADCC and CDC, and tandem anti-PD-L1 [218]. In trastuzumab-resistant preclinical models, this BsAb outperformed the two monoclonal antibodies alone, highlighting its potential to overcome trastuzumab resistance [218]. In a phase 1 clinical trial in patients with HER2+ solid tumors (NCT04162327), the HER2/PD-1 BsAb IBI315/Fidasimtamab showed no dose-limiting toxicity and a 20% objective response rate [219].

### 3.2. Immunotherapy-Enhancing ADCC and ADCP

Several HER2-targeted therapies are based on antibodies, and their effectiveness depends on ADCC and ADCP, as stated in the previous section. This fact prompts the study of the molecular mechanisms that control ADCC and ADCP to disclose regulators of these processes.

During HER2+ breast cancer treatment, ADCC is induced when the CD16 or FcƴRIII on the NK cells binds to the Fc domain of trastuzumab that has already recognized the HER2 molecule on the cancer cell. This interaction triggers the release of NK cells granules containing granzymes and perforins to the immunologic synapse, leading to tumor cell death. The NK cell effector function depends on the integration of activating and inhibitory signals triggered by cell surface receptors. Among the inhibitory receptors, NKG2A has been recognized as a mediator of tolerance and NK cell exhaustion [220], and it is also expressed in cytotoxic CD8+ T cells. Its ligand, HLA-E, a non-classical HLA class I, is frequently expressed in cancer cells [221]. With the purpose of neutralizing the inhibitory signal produced by the interaction of HLA-E and NKG2A, monalizumab, a humanized IgG4 monoclonal antibody against NKG2A, is now being evaluated in clinical trials [212]. The MIMOSA trial is a phase 2 study that expects to recruit 38 participants with metastatic breast cancer or locally incurable HER2+ breast cancer to be treated with monalizumab and trastuzumab.

On the other hand, ADCP is triggered in macrophages that encounter antibody-opsonized tumor cells. The binding of the FcƴRIII in macrophages to the Fc portion of the immunoglobulin attached to tumor cells induces its phagocytosis and intracellular death. It has been reported that tumor cells can express “don’t eat me” signals that act as innate immune checkpoints. One of these molecules is CD47, which interacts with the SIRPα receptor on macrophages and suppresses phagocytosis [222,223]. Recently, it has been demonstrated that administration of anti-CD47 antibodies significantly enhanced trastuzumab-mediated ADCP and improved antitumor responses and may help to resensitize refractory HER2+ breast cancers to trastuzumab treatment [224]. A phase 1 clinical study is currently recruiting 137 patients to study the safety of IMM2902, a HER2/SIRPα bispecific mAb-Trap antibody-receptor fusion protein, in patients with HER2-expressing advanced solid tumors (Table 1). In addition, B7-H4 was described as an immune checkpoint molecule in macrophages, defining a subpopulation of these cells that has immunosuppressive activity [225]. After ADCP, macrophages undergo a phenotypic change toward an immunosuppressive phenotype due to the upregulation of the immune checkpoint B7-H4. In preclinical models, blocking B7-H4 with a specific antibody synergized with trastuzumab’s antitumor effect [226]. Up to date, the only protocol in breast cancer testing antibodies against B7-H4 is the FPA150-001 clinical trial. This is a phase 1a/b clinical trial that studied the anti-B7-H4 antibody FPA150 as a monotherapy or in combination with pembrolizumab in patients with advanced solid tumors. Phase 1b recruited only HER2+ breast cancer patients (Table 1).

### 3.3. Immunotherapy Enhancing Adaptive Immune Response

The enhancement of the innate immune response can be achieved by tackling the inhibitory receptors described in the previous section. The strategy to enhance the adaptive immune response in HER2+ breast cancer is the use of activated antibodies.

One of them is utomilumab, a fully humanized IgG2 monoclonal antibody that binds to CD137 (4-1BB). CD137 is a costimulatory receptor rapidly expressed after antigen exposure on CD4+ and CD8+ T cells and transduces signals for proliferation, survival, and memory cell formation [227]. Two clinical trials are registered to treat HER2+ breast cancer patients with this antibody. The NCT03364348 trial was pursued to determine the recommended dose of utomilumab in combination with T-DM1 or trastuzumab in subjects with HER2+ advanced breast cancer. This is a phase 1 study that recruited 18 patients. Another trial called AVIATOR is a phase 2 study recruiting 100 patients aiming to determine the recommended dose of utomilumab in combination with T-DM1 or trastuzumab in subjects with advanced HER2+ breast cancer (Table 1). PRS-343, a BsAb that targets CD137 (4-1BB) and HER2, was developed, and preclinical data showed that activation of T cells unleashed tumor elimination even in trastuzumab-resistant models [228,229]. There is a phase 1 clinical trial to explore PRS-343 in HER2+ solid tumors (NCT03330561), and in another phase 1 clinical trial in HER2+ solid tumors, PRS-343 showed effectiveness alone or in combination with atezolizumab [230].

Another strategy is to ex vivo activate the T cells of each patient with anti-CD3 and anti-HER2 (trastuzumab) and then perfuse them in combination with pembrolizumab in women with metastatic breast cancer. There is a phase 1/2 study, NCT03272334, that estimates the enrolment of 33 patients.

Significance of tumor mutation burden, tumor-infiltrating lymphocytes, and PD-L1 expression in immunotherapy for HER2+ breast cancer

Tumor infiltrating lymphocytes (TILs), tumor mutation burden (TMB), and PD-L1 expression have all been proposed as biomarkers for patient selection for immunotherapy [231].

TMB was reported in 46 metastatic HER2+ breast cancer patients. The results showed that patients subjected to conventional HER2-directed treatments and chemotherapy, whose tumors have high TMB (˃100 somatic non-synonymous mutations), were associated with better overall survival than the ones that have low TMB [232].

Regarding TILs, the seminal work of Salgado in 2014 described a method for TIL quantification by pathologists’ analysis using H&E staining of breast cancer specimens [233]. This procedure was standardized and became available to any pathologist worldwide. The analysis of TILs in breast cancer reveals their association with positive outcomes in TNBC and HER2+ breast cancer. Both in early and advanced stages of HER2+ breast cancer, the presence of TILs is a predictor of a longer response to therapy [198,234,235]. In the case of the SortHER trial, the adjuvant administration of trastuzumab to primary HER2+ patients with 20% or more TILs showed that they can benefit from a shorter treatment. On the other hand, the CLEOPATRA study treated advanced and metastatic HER2+ patients with trastuzumab plus pertuzumab and chemotherapy and demonstrated that having 20% of TILs was a good prognostic. Moreover, the study proved that every 10% increase in TILs on the tumor core meant a longer overall survival [236,237]. Additional research found that low TIL levels in initial biopsies were associated with a low chance of achieving pCR in HER2+ breast cancer patients treated with trastuzumab plus pertuzumab and chemotherapy in the neoadjuvant setting [238,239].

Finally, it was demonstrated that the presence of TILs is highly correlated with PD-L1 expression in HER2+ and TNBC [240,241]. A study recruited 126 HER2+ patients who received neoadjuvant chemotherapy and trastuzumab to determine the clinical impact of PD-L1 expression. The results showed that 17.5% of the tumors expressed PD-L1 and were directly correlated with TILs and a better response to therapy [242]. Another study found that a cohort of 87 patients received neoadjuvant chemotherapy alone and 68 received neoadjuvant trastuzumab plus chemotherapy. In both cohorts, PD-L1 expression in TILs was an independent biomarker of pCR [243]. This evidence strongly supports the fact that high TILs and PD-L1 expression in HER2+ breast cancer are associated with a better response to adjuvant or neoadjuvant trastuzumab and/or pertuzumab treatment. However, a prospective clinical trial should be designed to validate the clinical significance of TILs and PD-L1 before including them in clinical practice as biomarkers of response to HER2-targeted therapies. PD-L1 expression in HER2+ breast cancer, on the other hand, has some caveats that support its use it as a biomarker of the success of anti-PD-1/PD-L1 antibody treatment in combination with HER2-targeted therapies, as stated in the section on PD-1 and PD-L1 antibodies, and more studies in that direction are granted. 

## 4. Cell Therapies

Cell-based immunotherapy consists of a personalized treatment that eradicates cancer cells by means of the patient’s own immune cells. Among cell therapies, living cells such as T cells, NK cells, and macrophages are used to treat cancer, taking advantage of the intrinsic ability of these cells to seek out, detect, and destroy foreign or abnormal cells in the body. These specialized immune cells are genetically engineered to recognize specific or selective tags on cancer cells isolated from the patient’s tumor, expanded ex vivo, and reinserted into the blood to boost the antitumor immune response.

To date, cellular therapies have been approved for liquid tumors, such as acute lymphoblastic leukemia (ALL) [244,245,246,247,248,249]. Yet, scarce progress has been made upon solid tumors for many reasons, such as aberrant vasculature formation that limits access of immune cells to the tumor core [250,251,252] and a dense extracellular matrix (ECM) that impairs infiltration, traps immune cells, and induces anergy [253,254,255,256,257]; the associated systemic inflammatory response syndrome (SIRS) [258,259,260,261] and off-target effects [262] represent some of the drawbacks that cell therapy encountered. In this section, we will address the current cell therapies available for HER2+ breast cancer.

### 4.1. CAR-T Cells

Tumor cells display different strategies to evade or disrupt T cell function, which may include the downregulation of canonic peptide-MHC presentation to decrease the chances of T cell detection and the immunosuppression of cell signaling through anti-inflammatory cytokine secretion or inhibition of co-receptor signaling. These strategies promote T cell exhaustion and dampen the antitumor immune response [263]. Chimeric antigen receptor (CAR)-T cells overcome many of these limitations by taking advantage of the antibody specificity and T cell cytotoxicity to directly target and destroy cancer cells. Although the first functional CAR-T cell was developed in 2002 by Maher et al. [264], it was not until 2017 that the FDA approved the CD19 CAR therapy to treat B cell malignancies [265]. However, the persistence of killing activity and the viability of CAR-T cells over time were important issues to address. In 2022, June et al. demonstrated that CAR-T cells persisted in two patients in remission who were infused with these cells 10 years earlier [266].

The basic structure of a CAR-T cell is divided into four main components based on their structure and function, including an extracellular target-binding domain, a hinge domain, a transmembrane domain, and one or more intracellular signaling domains [267]. The most common extracellular domain is composed of a scFv from the light and heavy chain variable regions of an antibody linked by the Gly4Ser peptide [268], the most common linker in CARs, but there are CAR-T cells that include a high-affinity TCR or a nanobody domain [269,270]. This domain is in charge of recognizing the tumor-specific antigens and activating the T cell, which is independent of MHC molecules. This therapy has bypassed the immune escape resulting from the downregulation of MHC molecules in tumor cells [271]. In turn, the extracellular domain is joined by the hinge region to the transmembrane domain, which provides stability to the receptor in the plasmatic membrane [272,273]. The transmembrane domain is bound to the intracellular one, which includes a primary signaling domain such as CD3ζ as well as co-stimulatory domains such as CD27, CD28, OX-40, ICOS, or CD137, which are required for full T cell activation, proliferation, and persistence [264,274,275,276]. When compared to previous CAR constructs, the fourth generation of folate receptor α -targeted CAR-T cells with three co-stimulatory domains, CD27, CD137, and CD28, demonstrated improved antitumor activity in breast cancer [277]. Another next generation CAR-T cell, called T cell redirected for antigen-unrestricted cytokine-initiated killing (TRUCK), includes a transgenic cytokine, such as IL-12, which encodes a secreted cytokine that is induced through NFAT signaling [278]. This cytokine release helps to support tumor killing and remains located in the tissue targeted by the TRUCK [278]. Moreover, the fifth generation of CAR-T cells is aimed at mitigating the adverse effects caused by the therapy itself, specifically neurotoxicity and CRS, which are the most common and negative aspects of CARs. This new CAR-T cell localizes cytokine signaling only in the presence of the cancer antigen, by including an intracellular signaling domain for cytokine receptors (ex. IL2-Rβ) that triggers the JAK/STAT pathway to induce effector function and receive the stimulation that would come under normal cytokine signaling [279,280]. Another strategy to improve CAR-T cells was to recognize two ligand-binding domains, so the T cell can distinguish between two different tumor antigens, either of which can activate the cell. In this way, tumor cell recognition, antitumor functionality, and antigen heterogeneity were improved [281], as demonstrated by researchers using a bivalent tandem CAR-T cell that targeted CD70 and B7-H3 in breast cancer [282]. Nowadays, there are about 500 therapies with CAR-T cells in clinical trials, but only six have been approved by the FDA for full use [283,284,285,286,287]. However, all of the approved CAR-T cell therapies are directed at blood cancers. Many of these CAR-T cell therapies in clinical trials target common neoantigens, such as EGFR or HER2.

Due to its high abundance in brain tumors, its known role in tumor progression, and the fact that both differentiated cells and cancer-initiating cells are eradicated by HER2-specific CAR-T cells [288], HER2 is a very interesting target antigen. However, the first patient treated with HER2-targeted CAR-T therapy did not survive the treatment as a result of CRS and off-target toxicity on the lungs, which caused respiratory distress and pulmonary edema [289]. Researchers improved tumor targeting by redesigning co-stimulatory domains and using the lower-affinity HER2-monoclonal antibody FRP5 to reduce the chances of CRS [290]. In spite of this, and although these CARs established safety, function, persistence, and expansion, they did not meet the standards to pass into the clinical setting. Ahmed and collaborators successfully administered 1 × 108 HER2-specific CAR-T cells with a CD28ζ endodomain without dose-limiting toxic effects in glioblastoma patients [291], while other preclinical data revealed that co-stimulation with CD137 displayed lower cytokine production and better anti-tumor efficacy compared to co-stimulation with CD28 [292].

Inspired by these results, two phase 1 clinical trials have started with optimized HER2-CARs for patients with HER2+ malignant glioma (NCT03389230) [293] and brain metastases of HER2+ breast cancer (NCT03696030) [294]. Another two clinical trials (NCT03500991 and NCT02442297) are investigating the locoregional delivery of HER2-specific CAR-T cells in pediatric central nervous system tumors that are HER2+, recurrent, and/or refractory [295,296]. Recently, a clinical trial that studies HER2-specific CAR-T vectors against ependymoma in children has started (NCT04903080), and results are awaited [297]. Tóth et al. designed in 2020 a murine T cell genetically modified to express a CAR that consists of a HER2 scFv derived from trastuzumab, a CD28 co-stimulatory endodomain, and a CD3ζ intracellular signaling domain [298]. The authors proved that these CAR-T cells effectively recognized and killed HER2+ tumor cells in vitro and significantly improved the immune response against human trastuzumab-resistant breast cancer even at very low numbers (only 7% of T cells were CAR-T cells), resulting in complete tumor remission within 57 days and a significant survival extension. The results confirmed that a strong antitumor effect against trastuzumab-resistant xenografts can be achieved with a small quantity of CAR-T cells [298], which could improve the above-discussed side effects of this therapy. Another advantage of CAR-T cells over antibody-based targeted therapies is their ability to penetrate the core region of the tumor and reach central tumor cells, which is a main challenge for monoclonal antibodies due to epitope masking by tumor cells and steric hindrance to their binding through matrix components [38,39,299,300]. At early stages of HER2+ breast cancer, treatment with monoclonal antibodies may be effective due to a less dense ECM, but as the tumor progresses, they become resistant due to a massive ECM that builds up and restricts access to antibodies [38,299,300]. However, while antibodies diffuse passively through the tumor nest, antibody-redirected immune cells can actively penetrate tumors. Immune cell therapy with adoptively transferred CAR-redirected T cells has been an engaging option to improve outcomes for patients with advanced or metastatic breast cancer [292,301,302]. In another study on trastuzumab-resistant HER2+ breast cancer, engineered T cells with a trastuzumab-derived HER2-specific CAR-T were obtained [303]. It binds to the same domain and thereby recognizes the same epitope as trastuzumab, with the scFv binding domain derived from trastuzumab (humanized 4D5), a stalk, the transmembrane and cytoplasmic regions of human CD28, and a CD3ζ signaling domain [303]. These T cells recognized de novo trastuzumab-resistant HER2+ breast cancer cells in three-dimensional cell cultures like spheroids and showed cytotoxic effects, while trastuzumab alone failed to do so. Moreover, it was proven that one dose of these CAR-T cells successfully eradicated established trastuzumab-resistant breast cancer xenografts that were not penetrated by trastuzumab to induce ADCC, resulting in long-term mouse survival [303]. These results demonstrate that CAR-T cells can effectively overcome the failure of antibody therapy due to epitope masking by steric impairment from the ECM components in HER2-positive breast cancer.

As approximately 50% of breast cancer brain metastases are HER2+, they pose another clinical challenge, in large part due to the inability of monoclonal antibodies to efficiently cross the blood–brain barrier. However, small-molecule HER2 inhibitors have been clinically approved for the treatment of breast-to-brain metastasis, but as single-agent therapies, they have shown no greater benefit [304,305] until the recent development of tucatinib, as described above. Researchers have been working on optimizing HER2-CAR-T cells for the treatment of brain metastasis in HER2+ breast cancer patients. Priceman et al. studied the intravenous, local intratumoral, and regional intraventricular routes of administration of these CAR-T cells in preclinical settings using human xenograft models of HER2+ breast cancer with brain metastases [292]. They found a strong in vivo antitumor effect after local intracranial administration of HER2-CAR-T cells in an orthotopic model of brain metastasis. Moreover, they demonstrated a robust antitumor effect when HER2-CAR-T cells were delivered intraventricularly into animals with multifocal brain metastases and leptomeningeal disease [294]. Intravenous delivery of HER2-CAR-T cells achieved only partial antitumor responses in mice, even at 10-fold higher doses compared with local or regional delivery to the brain. This led the authors to conclude that the intraventricular delivery of this therapy might hold promise for the treatment of multifocal brain metastases, while potentially limiting systemic T cell distribution and “on-target, off-tumor” effects and solving the difficulty of drugs breaking through the blood-brain barrier in tumor brain metastases [294]. An open-label phase 1 clinical trial (NCT03696030) was started in 2018 and is still recruiting patients with HER2+ breast cancer that has spread to the brain or leptomeninges and returned [294]. In this trial, the aim is to study the side effects and optimum dose of autologous HER2-targeted CAR-T cells, and although no results have been uploaded yet, the estimated completion time is August 2023. There are four other clinical trials regarding CAR-T cells and HER2+ breast cancer, all active and recruiting. One of them (NCT03740256) aims to evaluate combinatory strategies like oncolytic adenovirus plus autologous HER2-CAR-T cells to improve CARs recognition of tumor cells [306]. Another one (NCT04020575) focuses on HER2+ breast cancer patients with MUC1-positive tumors and advanced disease and studies the administration of a CAR-T cell (huMNC2-CAR44) that targets the extracellular domain of the cleaved form of MUC1 (called MUC1*), which is the isoform that acts as a growth factor receptor and is present on a large percentage of solid tumors, including breast tumors [307]. One positive aspect of this CAR-T cell is that the variable region only recognizes MUC1*, which is only present on tumor cells, and not MUC1, which is also present in healthy tissue, decreasing the “on-target, off-tumor” effects. A trial that began in 2020 investigates the safety and tolerability of CCT303-406 CAR-modified autologous T cells (NCT04511871) in patients with relapsed or refractory stage IV metastatic HER2+ solid tumors [308]. This CAR-T cell is a modified version of the HER2-CAR-T cells used in previous trials. The last clinical trial for HER2+ breast cancer patients that is open and recruiting is a phase 1/2 study (NCT04650451) that investigates the safety, tolerability, and clinical activity of HER2-specific dual-switch CAR-T cells, called BPX-603, administered with rimiducid, a lipid-permeable tacrolimus analogue, and a protein dimerizer, to subjects with previously treated, locally advanced, or metastatic solid tumors that are HER2+ [309]. The switch CAR-T (sCAR-T) cell has been developed as a programmable split system able to allow on/off control over cellular activity and also to enable a single CAR construct to target more than one antigen. This offers versatility to control the specificity and activity of engineered T cells to overcome heterogeneous tumors with improved safety [310]. The dual-switch HER2-CAR-T cell is comprised of a CAR that targets HER2 and a dual switch comprised of a chemical inducer of dimerization (CID)-inducible co-activation domain MyD88/CD40 (inducible MC; iMC), in which both the MyD88 and CD40 lack their extracellular domains, and an inducible caspase-9 safety switch, consisting of the CID-binding domain coupled to the signaling domain of caspase-9 [311]. When administered via CID technologies, rimiducid binds to the drug binding domain of iMC and activates both CD40- and MyD88-mediated signal transduction cascades [312]. This strategy has been shown to increase T cell proliferation, persistence, and resistance to T cell exhaustion, as well as upregulate immunomodulatory cytokines within the TME. As these T cells are engineered to only be fully activated by binding to both HER2 and rimiducid, their proliferation, activity, and toxicity can be controlled by adjusting the dose of rimiducid, preventing uncontrolled activation, which promotes the safety of the administered T cells [313]. Rimiducid is used to activate inducible caspase-9 produced by a modified gene included in this CAR T cell [313]. The activation produces rapid induction of apoptosis in activated modified T cells and resolution of the signs and symptoms of graft versus host disease (GVHD) within 24 h [314]. This aims to reduce the side effects caused by CAR-T cell administration. In other work, HER2 sCAR-T cells proved that the in vitro toxicity was specific against different HER2-expressing breast cancer cell lines (HER2 3+, 2+, and 1+, measured by immunohistochemistry) and that it was comparable to classical HER2-CAR-T cells [310]. It was also demonstrated that the tumor growth inhibitory effect of sCAR-T cells in NSG female nude mice bearing MDA-MB-453 tumors (HER2 +1 xenograft) was very similar to that obtained by infusion of classical HER2-CAR-T cells. These results confirm that sCAR-T cells are a better option than classical CAR-T cells since they work as sharply as the latter against HER2+ breast cancer cells and tumors but provide an additional tool to regulate activation and side effects [310].

CAR-T cells are also being studied in combination with other agents. One example is the inhibition of the epithelial-mesenchymal transition (EMT) in cancer cells as a possible strategy to enhance the activity of HER2-directed CAR-T cells [315]. In vitro studies have demonstrated that inhibition of the TGF-β1 pathway blocks the EMT process in cancer cells, restoring the cytotoxic activity of HER2-specific CAR-T cells [315]. Moreover, in the preclinical setting, researchers have found that TGF-β1 inhibitors exhibited promising activity in enhancing the killing capacity of HER2-CAR-T cells [315]. This makes TGF-β1 an attractive subject of study in relation to CAR-T cells. Other combinatorial strategies include ICI, such as anti-PD-1 molecules that have been tested both in vitro and in xenograft models. Evidence shows that the addition of an anti-PD1 antibody significantly improved the tumor-killing activity of HER2-specific CAR-T cells, since injected CAR-T cells were detected in the tumor stroma and delayed tumor development, and also augmented the apoptosis of tumor cells from 39% (CAR-T cells alone) to 49% (combined treatment), highlighting the potential of these molecules for improving cell-based therapies [316,317].

Regarding the recent description of HER2-low entities, CAR-T cells specific for them, have not been widely studied since these tumors were originally characterized as TNBCs or luminal tumors. A research group studied how to improve antigen loss or low antigen expression escape in HER2-expressing breast cancer. They developed a bispecific sCAR-T cell to target two antigens on HER2+ breast cancer cells: HER2 and IGF1R [310]. They tested their construct against a panel of HER2 3+, 2+, and 1+ breast cancer cell lines and found that it promoted more T cell activation, as measured by IL-2 production, against HER2 3+, 2+, and 1+ cells than sCAR-T cells and classical HER2-CAR-T cells but had no unexpected activation of T cells on HER2 0+ cells, highlighting its specificity [310]. Interestingly, they proved that although their bispecific sCAR-T cell had cytotoxicity effects comparable to monospecific HER2 sCAR-T cells in HER2 3+ and 2+ cells, bispecific sCAR-T cells exhibited the highest cytotoxicity rate against HER2-low expressing MDA-MB-231 cells (HER2 +1) compared to monospecific HER2 sCAR-T cells [310]. This suggests that the use of sCAR-T cells co-targeting IGF1R in addition to HER2 is a promising strategy for treating HER2+ breast cancers, showing enhanced cytotoxic activity and T cell activation in comparison to monospecific sCAR-T cells. Most importantly, these results mean a significant improvement against breast cancers with low HER2 expression, opening the possibility to overcome heterogeneous HER2 expression. Moreover, a research group has recently designed a modified CAR-T cell that is called “headless-CAR-T”, since it is missing the scFv region [318]. The design combines CAR-T’s potent intracellular signaling domains with a bispecific antibody arming strategy to redirect coactivated T cells’ non-MHC restricted cytotoxicity [318]. Particularly, in vitro assays proved that the HER2 hCAR-T with the co-stimulatory domain 41BBζ exhibited high levels of specific cytotoxicity directed at multiple tumor targets that express high and low antigen levels, like the HER2-low breast cancer cell line MDA-MB-231. In addition, based on the fact that novel anti-HER2 drugs like T-DXd are being used against HER2-low breast tumors and are being successful [105,319,320,321], in the near future CAR-T cells against TNBC patients may be tested in HER2-low subjects. Examples of TNBC targets included in CAR-T cells are: EGFR, ICAM1, MUC1, Mesothelin, CD133, CD44v6, CD70, TROP2, folate receptor α, and NKG2D, among others, and are extensively reviewed elsewhere [322]. Hopefully, in a few years, patients with low HER2 expression will benefit from the tireless effort of researchers and physicians to implement new technologies such as CARs to fight their tumors.

### 4.2. CAR-NK

Another immune cell type that is being studied for developing cancer immunotherapies is NK cells. In breast cancer patients, several factors regarding the TME account for NK cell dysfunction, which is mainly characterized by a decreased infiltration in the tumor core, increased death, altered metabolism and maturation, and reduced antitumor activity. Such dysfunction can be studied by looking at the NK cell effector molecules, like IFNγ, Fas ligand, CD107a, TRAIL, granzyme B, and perforin [323]. Not only the effector molecules are important when analyzing NK cell dysfunction; activating NK cell receptors is key as well. Examples of activating receptors that may be affected by the TME in breast cancer are NKp30, NKG2D, CD16, and DNAM-1 [323,324]. In addition, the accumulation of soluble molecules like IL-10 and TGF-β released by immunosuppressive cells can also impact the function of NK cells [324,325]. Moreover, it has been proven that L-kynurenine, a metabolite derived from IDO1, inhibits NK cell proliferation and cytotoxicity, and decreases the expression of NKp46 and NKG2D [326]. All of this evidence has drawn researchers’ attention to the need to build novel strategies to boost NK cell antitumor activity and increase their cytotoxicity. Based on the increasing proof that CAR-T cells have been a successful therapeutic tool, genetic modification of NK cells is the next step [327].

Through their antibody Fc receptor, FcγFIII or CD16, NK cells can trigger an ADCC against cancer cells [328]. Although NK cells can recognize cancer cells by antigen-independent means (such as downregulation of HLA or damage-associated molecular patterns on the cancer cell membrane), CAR-NK cells are designed to additionally target tumor cells by antigen-dependent means, through the stimulation of their receptor [329,330,331,332,333,334]. This bimodal approach for targeting cancer cells by both antigen-independent and -dependent mechanisms plays a crucial role in heterogeneous tumors, where there is an increased variability in tumor antigen expression and tumor escape is more likely to take place [335]. Since CAR-NK cells are derived from CAR-T cells, they share the same intracellular signaling domains (CD3ζ, CD28, CD137). Recent studies in ALL, osteosarcoma, and prostate cancer have proved that replacing those T cell-specific signaling domains with receptor domains associated with NK cell signaling, like DAP10 and DAP12, revealed superior antitumor potential in primary NK cells or the NK-92 cell line, with respect to CAR-NK cells bearing T-specific internal signaling domains [336,337,338]. Moreover, adding CD244 as a CAR-NK cell so-stimulatory domain provided rapid NK cell proliferation and expansion, enhanced cytotoxicity, and solid antitumor properties [339].

CAR-NK cells pose several advantages over CAR-T cells, since adoptive cell transfers do not induce GVHD and HLA matching is not required. Moreover, CAR-NK cells represent a safer option in terms of side effects, as they have been shown to have no neurotoxicity or CRS events [340,341,342]. This is related to the fact that activated NK cells secrete mainly IFNγ and GM-CSF, while CAR-T cells secrete several pro-inflammatory cytokines, such as IL-1, IL-2, IL-6, TNFα, IL-8, IL-10, and IL-15, which are more prone to trigger CRS [343,344]. Also, some studies proved that a subset of memory-like NK cells developed from CAR-NK cells encountering tumor cells, which may provide additional benefits regarding these CARs and rationale for future strategies of combining CARs with other immunotherapy approaches [345,346,347]. Furthermore, NK cells can be obtained from autologous or allogeneic sources, such as umbilical cord blood (UCB), human embryonic stem cells (ESCs), induced pluripotent stem cells (iPSCs), peripheral blood (PB), and clonal NK cell lines. Each source has advantages and disadvantages; for example, PB-NK cells exhibit high expression of activating factors with the potential to destroy tumor cells (such as CD16, NKG2D, and NKp44), but collecting them from peripheral blood is expensive and time-consuming because NK cells represent only 10–15% of all lymphocytes, and subsequent methods must be used to expand them ex vivo [348,349,350]. UCB-NK cells are easier to collect, they exhibit an increased proliferative capacity compared to PB-NK cells, and they pose a lower risk of GVHD. They are immature, with low expression of activating receptors such as NKp46, NKG2C, IL-2R, DNAM-1, and CD57 and high expression of the inhibitory molecule NKG2A [351,352,353,354]. For researchers, this implies activating, stimulating, and expanding them ex vivo as well [355]. The iPSCs source provides NK cells with a high proliferative capacity and cytotoxic activity, as well as homogeneous expression of activating molecules and receptors, such as NKG2D, NKp46, Fas, and TRAIL, and poor expression of inhibitory molecules [356,357,358]. Clonal NK cell lines, such as NK-92, NK-YS, NKL, and NKG, are eligible candidates to obtain homogeneous NK cells since they are easy to expand and have solid antitumor properties. The most common cell line used to generate allogeneic NK cells is the NK-92 cell line, since it generates more perforin, granzyme B, and other cytotoxic cytokines than the others. However, they fail to express key activating receptors such as CD16, NKp46, and NKp44, which compromises their cytotoxic performance [359,360,361,362]. In addition, clonal NK cell lines must be irradiated before being infused into the patient to avoid permanent allogeneic tumor engraftment, which reduces NK cell persistence in the host [363]. NK cells derived from iPSCs are proven to expand, persist, and kill tumor cells efficiently within the TME and are easily manipulated to obtain genetically modified cells, which led to the creation of CAR-NK cells derived from iPSCs [364]. These CAR-NK cells presented increased life span and target specificity; they were also more resistant to exhaustion, and they were able to activate other immune cells to efficiently kill tumor cells [364]. As stated before, the main drawback of using NK cells for CARs is that these cells tend not to expand once transplanted back into the patient, which imposes the need to be expanded ex vivo before reinfusion to be able to migrate and penetrate into solid tumors [365,366]. This limitation is being addressed by designing NK cells with transgenes that constitutively secrete or express cytokines on their membrane to support cytokine supply while controlling their side effects [367,368,369]. Today, there are more than 30 CAR-NK cell clinical trials, most of which are directed at blood cancers [370,371,372], but as with CAR-T cells, this therapy needs to overcome the challenge of tumor infiltration to achieve higher response rates.

The first report of CAR-NK cells as a therapy for breast cancer dates back to the development of a CAR-NK cell targeting HER2 using NK-92-cvFv(FRP5)-zeta cells [373]. They reported that intravenous injection of their construct in mice bearing the HER2+ NIH 3T3 tumor reduced cancer progression after 12 and 24 h of injection [373]. More recently, a humanized CAR-NK cell using NK-92-cvFv(FRP5)-zeta cells based on the specific anti-HER2 antibody FRP5 and carrying the CD3ζ and CD28 co-stimulatory domains (CAR-NK cell 5.28.z) was obtained. This CAR-NK cell proved to effectively lyse HER2-expressing MDA-MB453 breast cancer cells in vitro [374]. What is more, this lysing capacity was retained even after the irradiation of CAR-NK 5.28.z cells as a safety measure to prevent tumor engraftment. Furthermore, not only was this CAR-NK selectively enriched in orthotopic breast carcinoma xenografts of MDA-MB-453 cells, but they also retained their specific recognition of HER2+ tumor cells and antitumor activity observed in in vitro studies [375]. This CAR-NK cell was also used in other strategies to demonstrate that retargeting NK cells with CAR technology is an effective way to overcome the intrinsic resistance of the HER2+ breast cancer cell line MDA-MB-453 to NK cell ADCC [375]. These findings are crucial for HER2-low tumors, which exhibit few HER2 molecules in their membranes or have a heterogeneous receptor expression.

NKG2D is an activating receptor of NK cells that is widely expressed in several tumors and can be used as a target for adoptive immunotherapy. However, shedding or downregulation of NKG2D ligands (NKG2DL) can prevent NKG2D activation, allowing cancer cells to evade NKG2D-dependent immune surveillance [376]. Based on this fact, a bispecific antibody that targets NKG2D and HER2 (NKAB-ErbB2) to specifically attack tumor cells expressing NKG2D independently of membrane-anchored NKG2DLs was designed. Simultaneous treatment of NKAB-ErbB2 with the NKAR-NK-92 cell resulted in effective lysis of MDA-MB-453 and JIMT-1 cells in vitro, accompanied by an elevated production of IFNγ, and showed no effect in MDA-MB-468 cells, proving the specificity of the system against HER2 [376]. Moreover, although the in vivo efficacy of the construct was not tested in breast cancer, it proved to effectively inhibit the growth of HER2+ tumors, resulting in treatment-induced endogenous antitumor immunity and cures in most animals [376]. These studies highlight the possibility that CAR-NK cells may be a powerful source of immunotherapy in the context of HER2+ breast cancer. Regarding clinical trials with CAR-NK cells, to date, there is no trial specific for HER2+ breast cancer patients. However, there is one clinical trial (NCT03383978) that is actively recruiting and aims to study the intracranial injection of NK-92/5.28.z cells in combination with intravenous ezabenlimab in patients with recurrent HER2+ glioblastoma [377]. Results are expected for October 2023, but still no data has been disclosed. Two clinical trials on a variety of solid tumors, including breast (NCT05528341 and NCT05137275), are also studying CAR-NK cells [378,379]. The first one analyzes NKG2D-CAR-NK92 cell infusions in the treatment of relapsed or refractory solid tumors, such as breast, lung, gastric, ovarian, cervical, renal, malignant melanoma, and osteosarcoma. The second one studies an allogeneic CAR-NK cell conjugated to an antibody against trophoblast glycoprotein (5T4) called CAR-raNK in patients with locally advanced or metastatic solid tumors [380]. Although CAR-NK cells seem to be very promising, optimizing a procedure for the expansion and activation of harvested NK cells is required to ensure a homogeneous population that can produce memory-like, unexhausted NK cells. Additionally, the utility of CAR-NK cells in the treatment of breast cancer may need further modifications of the NK cells beyond CAR transduction to increase trafficking and desensitize them to the immunosuppressive TME.

### 4.3. CAR-M

A major obstacle to CAR-T and CAR-NK cell therapies to treat solid tumors is the difficulty of infiltration by these genetically modified immune cells due to the stromal ECM that surrounds many solid tumors [381]. Although most macrophages originate in the bone marrow and circulate through peripheral blood until a cytokine gradient attracts them to the tumor bed [382], tissue-resident macrophages have been in the spotlight for cancer immunotherapy. According to cytokine stimulation in the TME, resident macrophages may polarize to the M1-like antitumor and pro-inflammatory subtype (through IFNγ and TNFα or recognition of LPS) that activates the adaptive immune system [383,384,385,386] or to the M2-like protumoral and immunoregulatory subtype (through IL-4, IL-10, IL-13, and TGF-β), which promote angiogenesis and tissue repair and are strongly associated with tumor survival and progression [387,388,389,390]. However, both M1-like and M2-like macrophages may switch between polarization states when exposed to opposing stimuli [383]. Usually, when macrophages from peripheral blood infiltrate solid tumors, they are influenced by cytokines such as TGF-β and IL-10 and turned into TAMs [390], which resemble the M2-like polarization state [391,392,393].

Given the plasticity, different strategies are being studied to enhance macrophage anti-tumor activity and inhibit the development of immunosuppressive macrophages in order to increase M1-like macrophages and decrease TAMs. Among those strategies, it is worth mentioning the inhibition of the CCL2-CCL2R axis [393,394,395,396], enhancing the activation of macrophages and dendritic cells through TLR and CD40 agonists [397], inhibiting the immunosuppressive pathway of CD47-SIRPα [397], and promoting reprogramming to the M1-like subtype by inhibiting the PI3Kγ pathway [398,399], MARCO signaling [400], class I and II histone deacetylases [401] or activating the CD11b [402]. Researchers have put their focus on genetically modified macrophages since they have a natural ability to infiltrate tumors and, with the appropriate modifications, are able to maintain their anti-tumoral profile [403]. The first CAR macrophage cell was designed to direct phagocytosis activity toward tumor cells and was named CAR-P. It consisted of a scFv that targeted CD19 or CD22 and an intracellular signaling domain that was a phagocytic receptor like Megf10 or FcRγ [403]. The authors proved low phagocytosis rates against CD19+ Raji cells, which were improved when a tandem PI3K recruitment domain was added to the CAR-P, together with anti-CD47 antibodies [403].

A CAR-M was developed by using an adenoviral vector in primary human macrophages that were able to sustain the M1 phenotype. The CAR-M structure was a CD3ζ-based CAR with a scFV to target HER2, and they found that the HER2-CAR-M cells exhibited antigen-specific phagocytosis of HER2+ beads and SKOV3, a HER2+ ovarian cancer cell line, in a dose- and time-dependent manner [404], and that cells from 20 representative normal tissues were not phagocytosed. The administration of a single intravenous injection of HER2-CAR-M cells in animals significantly reduced the tumor burden of SKOV3 lung metastasis and intraperitoneal carcinomatosis, increased overall survival, and decreased lung metastases compared to the control group in the first approach. Characterization of the CAR-M activity showed that they expressed pro-inflammatory cytokines and chemokines, converted bystander M2 macrophages to M1, upregulated antigen presentation machinery, recruited and presented antigen to T cells, and resisted the effects of immunosuppressive cytokines [404]. In humanized mouse models, HER2-CAR-M cells induced a pro-inflammatory TME and boosted anti-tumor T cell activity by recruiting T cells and cross-presenting antigens from phagocytosed cells. Furthermore, a murine HER2-CAR-M cell was demonstrated to effectively target and kill the HER2+ AU-565 breast cancer cell line in a dose-dependent manner [405]. Moreover, it was able to induce MHC-I and MHC-II expression on tumor cells and cross-present tumor-associated antigens, causing the activation of CD8+ T cells. It has been demonstrated that using HER2-CAR-M to treat CT26-HER2+ tumor-bearing mice increases intratumoral T and NK cell infiltration, dendritic cell infiltration and activation, and TIL activation. Furthermore, HER2-CAR-M administered locally in these tumors inhibited the growth of contralateral HER2− tumors while also preventing antigen-negative relapse upon HER2− tumor rechallenge, indicating epitope spreading and the induction of long-term immune memory [405]. What is more, since their CT26 HER2+ tumor model is resistant to anti-PD-1 therapy, they combined HER2-CAR-M with anti-PD-1 antibodies and found that the combination further improved tumor control and overall survival [405]. Indeed, the HER2-CAR-M CT-0508 is under evaluation in a phase 1 clinical trial for patients with relapsed/refractory -verexpressing solid tumors (NCT04660929), including breast cancer [406]. Another clinical trial (NCT05007379) that studies HER2-CAR-M is being conducted in breast tumor samples to develop patients’ derived organoids and test the antitumor activity of newly developed HER2-CAR-M [407]. The first outcome measure will be the comparison of the HER2-CAR-M’s antitumor activity against organoids from HER2-, HER2-low, and HER2+ breast cancers, as well as non-modified macrophages. A secondary outcome measure will be a comparison of the HER2-CAR-M’s antitumor activity against organoids from early and advanced breast cancer patients.

One of the main barriers for cell therapy has been the ECM, as mentioned before. However, it is known that metalloproteinases (MMPs) can degrade tumor ECM and that macrophages are key producers of MMPs [408]. This fact has motivated researchers to use macrophages as means to degrade the tumor ECM by producing specific MMPs as strategies to overcome resistance to cellular-based therapies. One example is the design of a CAR targeting HER2 for macrophages with the hope of activating MMPs to degrade the matrix and broaden the path for T cells entry into the HER2+ 4T1 mouse tumor [409]. This HER2-CAR-M cell was aimed at activating CD147, which is part of the signaling pathway to activate MMPs. CAR-147 cell infusion significantly inhibited tumor growth, reduced tumor collagen deposition, and promoted T cell infiltration into the tumor core in 4T1 tumor-bearing mice [409]. What is more, in order to study the cytotoxic effects of CARs related to CRS, the analysis of the peripheral blood of CAR147-infused mice revealed that the inflammatory cytokines TNF-α and IL-6, which are key factors in CRS, were significantly decreased. This implies that using this CAR construct is safe, and that targeting the ECM of breast tumors with CAR-M could be an interesting approach for developing novel therapeutic options in this setting [409].

CAR-Ms has demonstrated clear superiority compared to other CARs in the preclinical treatment of solid tumors. This can be due to the advantages that macrophages have over T cells and NK cells [386], for example: (1) CAR-Ms are able to enter solid tumors since they have the inherent tumor-homing ability of myeloid cells; (2) CAR-Ms kill tumor cells that express the antigen by phagocytosis or secretion of cytokines; (3) CAR-Ms promote an inflammatory TME by promoting recruitment of T cells and other immune cells and by secreting pro-inflammatory cytokines and chemokines; (4) CAR-Ms are able to counteract the immunosuppressive TME; and (5) CAR-Ms are antigen-presenting cells and are able to present tumor associated antigens to T cells and induce adaptive immune responses.

It is evident that the compelling body of evidence presented here has demonstrated the feasibility of using CAR-immune cells as a therapeutic product to overcome therapy resistance and personalize patients’ treatments. However, researchers need to keep digging to better understand the mechanisms behind their function, which will allow the design of efficient strategies to manipulate them and improve their present flaws.

## 5. Anti-Cancer Vaccines

Since the disclosure of breast cancer as immunogenic and the success of the Sipuleucel-T vaccine in the treatment of prostate cancer, it has opened the door to thinking about vaccination as a new therapeutic modality in HER2+ breast cancer [410]. Vaccines have a number of advantages compared to standard cancer treatments like chemotherapy and anti-HER2-immunotherapy via monoclonal antibodies: better tolerance, lower toxicity, and a long-lasting immune response with tumor specificity [411]. Possibly, cancer relapse can be avoided by activating long-term immunological memory with an effective vaccine that can protect against various tumor antigens. What is more, vaccines do not need to be administered frequently [412]. There are preclinical studies in progress, and some candidate vaccines to treat HER2+ breast cancer are currently in clinical trials. However, so far, the FDA has not approved any vaccines for breast cancer treatment. To develop an effective and safe HER2+ breast cancer vaccine, researchers must first better understand the tumor microenvironment, identify specific tumor-associated antigens, develop effective vaccine formulations, and determine the best route of administration [413].

As mentioned previously, therapeutic resistance is common when using HER2-targeted therapies and it leads to disease progression and recurrence [414]. A novel vaccine was developed using an alpha viral vector encoding a portion of HER2 (VRP-HER2), and it was tested in a phase 1 clinical trial, where they demonstrated that it was well tolerated with no dose-limiting toxicity [415]. Preclinical models additionally demonstrated the immunogenicity and antitumor activity of the vaccine. Pre- and postvaccination analysis of peripheral blood mononuclear cells revealed the expansion of a HER2-expressing memory CD8+ T cell population in a subset of patients and an increase in the presence of HER2-specific polyfunctional antibodies. The presence of an expanded HER2-specific memory T cell population was associated with improved progression-free survival in this study. These results support further trials combining this vaccine with ICIs such as anti-PD1 to better engage expanded T cell populations [415]. A phase 2 study, in which subjects will receive the VRP-HER2 immunizations plus pembrolizumab (NCT03632941), is being conducted.

Other authors have proposed that cancer vaccines have had limited success because clinical trials have focused on later stages of the disease when tumor burden is at its maximum and even standard therapies, at times, have minimal benefit [416,417]. With respect to breast cancer and potential intervention with a cancer vaccine, ductal carcinoma in situ (DCIS) is an ideal stage to test it. DCIS is an early pre-invasive breast malignancy that acts as a bridge between normal breast tissue and invasive breast cancer; therefore, implementing a breast cancer vaccine in the early stages of the disease, such as DCIS, may allow for better results because the tumor burden is low [418]. Therefore, a proposed vaccine was tested at this stage of the disease. The protocol consisted of activating dendritic cells (DCs) with cytokines and a clinical-grade bacterial LPS to induce a unique battery of chemokines and cytokines, such as IL-12, which can condition T cells for superior antitumor immunity and memory (NCT02063724) [419,420,421]. The DCs are pulsed with synthetic peptides based on the HER2/neu sequence that are capable of sensitizing helper T cells in most individuals [422,423], and Th1 responses are associated with good clinical outcomes in breast cancer [424]. The DCs were administered to patients with DCIS in the neoadjuvant setting in an effort to attack the early stage of the disease. This vaccine was successful in inducing strong, long-lasting immune responses and either reducing or eliminating HER2/neu expression in most vaccinated patients [425]. This promising strategy could be further studied for treating early breast cancer, preventing recurrence, or for total prevention of primary disease in high-risk populations [426].

Another approach that is currently being tested in a phase 1 clinical trial is plasmid DNA vaccine encoding the HER2 intracellular domain in advanced HER2+ breast cancer patients. Patients with HER2+ breast cancer who are treated with trastuzumab develop high levels of HER2-specific Th1 in their peripheral blood and exhibit a better clinical outcome. However, only a minority of patients develop measurable anti-HER2 immunity after treatment. Therefore, vaccines designed to increase HER2-specific T-helper cells could induce HER2 immunity in a majority of patients to achieve a clinical response [427]. One reason for HER2-directed vaccines failure may be the construct of the vaccine, as HER2 vaccines designed to stimulate CD8+ T cells using short epitopes may restrict response to a limited number of HLA or short-lived immunity [428]. Protein-based vaccines stimulate high levels of antibodies, which require Th2 T cells, potentially inhibiting Th1 and cytotoxic T cell proliferation [429]. Encoding the complete intracellular domain (ICD) of HER2 into a plasmid allows for delivery of many more T-helper epitopes and the potential for persistence of the plasmid in the skin, which may influence the development of T cell memory [430]. Disis et al. demonstrated that immunization with 100 μg of the HER2 ICD plasmid-based vaccine was associated with the generation of HER2-specific Th1 cells in most patients with HER2-expressing breast cancer, and it is currently being evaluated in a randomized phase 2 trial [430].

Finally, it is worth mentioning a study carried out in patients with HER2-low tumors that, although it did not turn out as expected, can provide some relevant data. It has been shown that passive immunotherapy or monoclonal antibodies are not effective in patients with HER2-low tumors; therefore, we can assume that active immunotherapy or HER2-targeted vaccines could be beneficial for these patients. Based on this assumption, a HER2-derived peptide vaccine, including nelipepimut-S (NPS), was developed and combined with granulocyte-macrophage colony-stimulating factor (GM-CSF) as an immunoadjuvant. Preclinical data provide evidence for synergism between HER2-targeted peptide vaccines and trastuzumab [431,432,433]. This study is the first randomized trial evaluating the combination of trastuzumab with vaccination in the adjuvant setting. The combination of both therapies is safe, but unfortunately, in patients with HER2-low breast cancer, no significant difference in disease-free survival was observed in the intention-to-treat analysis; however, a significant clinical benefit was observed in patients with TNBC (NCT02297698). These data should alleviate concerns about combining active immunotherapy with monoclonal antibodies targeting HER2 [434].

As we mentioned at the beginning of the section, even though active vaccination therapy for HER2+ breast cancer has several theoretical advantages, clinical trials evaluating breast cancer vaccines have provided limited evidence of clinical benefits despite the successful induction of immune responses [435]. This could be explained by the fact that effective antitumor immunity that is stimulated by a vaccine is not long-lasting enough to produce significant survival benefits. The reason why the antitumor immune response is not durable enough can be attributed to different reasons: suboptimal vaccine formulations, immune tolerance developed for specific tumor antigens, and the immunosuppressive microenvironment. This implies further investigation into the optimal dose and immunization schedule, routes of administration, and options for immune boosters, in order to achieve an efficient vaccine. Booster inoculations were shown to be able to maintain immunity, and those who received scheduled booster inoculations were less likely to recur [435,436].

Immunotherapy is defined as the use of materials that increase and/or restore the ability of the immune system to prevent and fight diseases [437]. In this aspect, the concept behind cancer vaccines—that the autologous immune system can be mobilized to fight cancers—has never been abandoned. Although the results of vaccine clinical trials against HER2+ are still not satisfactory, the generation of new strategies could eventually lead to greater success for the proposed therapies.

## 6. Exosomes

For many years, exosomes were considered discarded, non-functional cellular components. However, over the years, exosomes have been shown to be an important tool for intercellular material and information exchange [438]. The exosome is a nanoscale cell-secreted membrane vesicle with a diameter of 30 to 150 nm that participates in intercellular communication by delivering lipids and nucleic acids to the receptor cells. They are involved in communication between neighboring cells or are transported to distant tissues and organs where they send signals or messages to receptor cells [439]. It has also been shown that exosomes can reverse drug resistance among heterogeneous populations of cancer cells, ultimately influencing the efficacy of treatment for many types of cancer [440]. Therefore, the development of new exosome-based therapies to treat HER2+ breast cancer could be very useful.

As mentioned in the vaccines section, HER2 vaccines using HER2-specific peptides, proteins, DNA, or DCs showed relatively limited antitumor effects. Combining vaccines with exosomes could be an interesting and novel approach to treat HER2+ patients. Exosomes (EXOs) released by DCs are known to be enriched in immunological molecules important for DC’s stimulatory machinery [441]. Wang et al. developed HER2-specific EXO-T vaccines (HER2-Texo) using active polyclonal CD4+ T cells with uptake of HER2-specific DC-released EXOs. They showed that HER2-specific EXO-T-stimulated cytotoxic T lymphocytes (CTLs) showed potent therapeutic effect against both T47D, an HER2+ breast cancer cell line, and BT474, a trastuzumab-resistant HER2+ breast cancer cell line, in athymic nude mice [442]. Furthermore, the EXO-T vaccine was able to convert CTL exhaustion into chronic infection via CD4+ T cell CD40L signaling-induced activation of the mTORC1 pathway, resulting in CTL proliferation, IFN-γ production, and CTL cytotoxicity rescue [443]. The tumor-specific effector CTLs, after fighting the tumor for a while, end up exhausted and lose their function [444]. The novelty of this new EXO-T vaccine is that it could reverse the exhausted phenotype of CTLs. The long-term goal is to develop a human therapeutic HER2/Neu-specific EXO-T vaccine using autologous polyclonal T cells with uptake of HER2/Neu-specific autologous DC (DCHER2/Neu)-released EXOs as a novel personalized vaccine for breast cancer [445].

Another interesting approach is the one proposed by Han et al., in which a vital autophagy-regulating miRNA (miR-567) that blocks autophagy-promoting cell sensitivity to trastuzumab in HER2+ breast cancer cells was studied. Autophagy is associated with the maintenance of tumor cell survival under adverse conditions, including nutrient deficiency, chemotherapy, and radiation treatment [446]. Trastuzumab-resistant cells showed increased autophagic activity. ATG5 protein is involved in the early stages of autophagosome formation [447], and miR-567 regulates the expression of ATG5 at the post-transcriptional level. Han et al. discovered that miR-567 levels were lower in trastuzumab-resistant patients versus responding patients. Furthermore, in vitro assays showed that miR-567 was also downregulated in trastuzumab-resistant cells. Overexpression of miR-567 reversed chemoresistance, whereas silencing of miR-567 induced trastuzumab resistance, both in vitro and in vivo. As a result, miR-567 could be packaged into exosomes and delivered to recipient cells, suppressing autophagy and reversing chemoresistance by targeting ATG5. Exosomal miR-567 could be a promising therapeutic target and prognostic indicator for HER2+ breast cancer patients since it plays a key role in reversing trastuzumab resistance through autophagy regulation [448].

Another property of exosomes that can be exploited is that they are a good option for gene targeting due to their nature being non-toxic, non-immunogenic, biodegradable, and orientable [449,450]. By engineering exosome-producing cells, ligands can be expressed and fused with exosomal surface proteins to target cancer cell receptors. In one study, HER2+ breast cancer cells were targeted with a modified exosome producing engineered HEK293T cells transduced with a lentiviral vector bearing the LAMP2b-DARPin G3 chimeric gene. Stable cells expressing the fusion protein were selected, and the exosomes produced by these cells were isolated and then loaded with short interference RNAs for subsequent delivery to the SK-BR-3 cells. Stable HEK293T cells produced DARPin G3 on the surface of exosomes, which can bind specifically to HER2 and deliver short interference RNA molecules against TPD52 into the SK-BR-3 cell line, thereby downregulating gene expression [451]. This study represents a new option to facilitate gene therapy and drug delivery to tumor cells, providing an additional option for gene therapy and drug delivery.

## 7. Conclusions

Since the description of the HER2 receptor as a biomarker and an attractive therapeutic target, several drugs have been developed, with trastuzumab dominating the treatment landscape for this breast cancer subtype. However, resistance events impair the clinical benefit, indicating that the development of novel HER2-targeted therapies is not only desirable but also required. In this sense, there are more than 2000 clinical trials registered to date to evaluate new HER2-targeted therapies. Along with drug development, new tools, such as single-cell sequencing, theranostics, spatial transcriptomics, and proteomics, have been developed in tandem with the technological advances [452,453,454,455,456]. The HER2+ landscape treatment can count on multiple HER2-targeting monoclonal antibodies, HER2-targeted ADCs, which have proven to be promising in the clinical setting; for example, T-DXd became the second line of treatment for HER2+ breast cancer patients. Several new TKIs have been developed and tested, among others, that have improved the management of HER2+ breast cancer patients and will change clinical practice. One of the main complications in cancer is the establishment of metastasis, and in this sense, some of the therapies mentioned in this manuscript have shown to be effective in the metastatic setting. All the above-mentioned initiatives encourage the scientific community to collaborate on the development of new HER2-targeted therapies and clinical trials testing different treatment combinations that could overcome tumor progression or even metastasis. The importance of discovering new biomarkers that can predict therapy response must be emphasized in this regard, as this will allow us to determine which patients will benefit from which therapies or combination treatments offer them the best treatment option. 

## Figures and Tables

**Figure 1 cancers-15-01987-f001:**
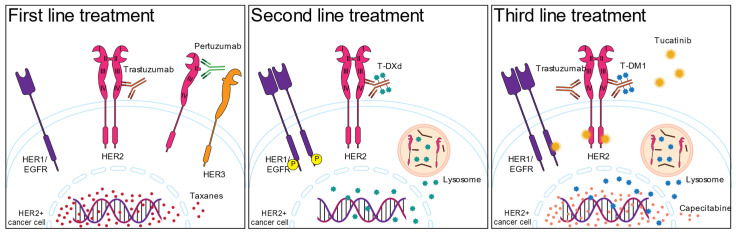
The current standard of care for HER2+ breast cancer treatment. The numbers on the HER2 molecule represent the protein domains.

**Table 4 cancers-15-01987-t004:** Clinical trials combining immunotherapies with HER2-targeted therapies.

ID	Type of Study	Status	No. Patients	Population	Treatment
**Pembrolizumab (anti PD-1 antibody)**					
NCT02129556 PANACEA	Phase 1/2 Single arm	Completed	58	Metastatic HER2+ breast cancer, trastuzumab-resistant	Pembrolizumab with trastuzumab [205]
NCT03747120	Phase 2 open-label, randomized	Recruiting	174	Naive patients with invasive human HER2+ breast cancer whose primary tumors are > 2 cm and/or clinically lymph node-positive	Neoadjuvant trastuzumab, pertuzumab, and paclitaxel Arm A: trastuzumab + pertuzumab + paclitaxel, Arm B: trastuzumab + pertuzumab + paclitaxel+ pembrolizumab or Arm C: trastuzumab + pembrolizumab + paclitaxel [206].
NCT03032107	Phase 1b	Active, not recruiting	27	Metastatic HER2+ breast cancer	Pembrolizumab + T-DM1
NCT04789096 TUGETHER	Two arms, phase 2	Not yet recruiting	50	Women or men with HER2+, metastatic breast cancer, who have progressed since previous treatment	Pembrolizumab + tucatinib + trastuzumab (PD-L1+) or Pembrolizumab + tucatinib + trastuzumab + capecitabine (PD-L1-)
NCT04660929	Phase 1, open label	Recruiting	48	HER2+ recurrent or metastatic solid tumors	Anti-HER2 CAR macrophages + pembrolizumab
NCT05020860 I-SPY trial	Phase 2, open label	Not yet recruiting	185	Early HER2+ breast cancer	Neoadjuvant paclitaxel + trastuzumab + pertuzumab in combination with pembrolizumab
NCT03272334 Breast-47	Phase 1/2	Recruiting	33	Metastatic HER2+ breast cancer	Pembrolizumab administered in combination with HER2 and CD3 bispecific antibody armed activated T cell (BATs) infusions
**Atezolizumab (anti-PD-L1 antibody)**					
NCT02924883 KATE2	Phase 2, double blind	Completed	133	Locally advaced or metastatic HER2+ breast cancer	Atezolizumab and trastuzumab-emtansine (T-DM1) Arm 1: T-DM1 + atezolizumab, Arm 2: T-DM1 + placebo [207]
NCT04740918 KATE3	Phase 3, doble blind	Recruiting	320	Locally advanced or metastatic HER2+ and PD-L1+ breast cancer who have received prior trastuzumab- (+/− pertuzumab) and taxane-based therapies	Atezolizumab and T-DM1 Arm A: T-DM1 + placebo, Arm B: T-DM1 + atezolizumab
NCT03726879 IMpassion050	Phase 3, doble blind	Active, not recruiting	454	High-risk early HER2+ breast cancer	Atezolizumab or placebo in combination with neoadjuvant doxorubicin + cyclophosphamide followed by paclitaxel + trastuzumab + pertuzumab (ddAC-PacHP) Arm 1: Atezolizumab + ddAC-PacHP. Arm 2: placebo + ddAC-PacHP [208]
NCT04873362 Astefania	Phase 3, doble blind	Recruiting	1700	High risk HER2+ breast cancer following preoperative therapy	Adjuvant atezolizumab or placebo and T-DM1. Arm A: placebo + T-DM1. Arm B: Atezolizumab + T-DM1 [209]
NCT02605915	Phase 1, open label	Completed	98	HER2+ and HER2− breast cancer	Atezolizumab + T-DM1 or with trastuzumab and pertuzumab (with and without docetaxel) in patients with HER2+ breast cancer and atezolizumab + doxorubicin and cyclophosphamide in HER2− breast cancer
NCT03417544	Phase 2	Active, not recruiting	33	Central nervous system metastases in patients with HER2+ breast cancer	Atezolizumab + pertuzumab + high-dose trastuzumab
NCT03199885	Phase 3, doble blind	Active, not recruiting	600	First-line metastatic HER2+ breast cancer	Arm I: pertuzumab + trastuzumab + taxane therapy + atezolizumab. Arm II: pertuzumab + trastuzumab + taxane therapy + placebo
NCT04759248 ATREZZO	Phase 2, open label	Recruiting	110	Advanced/metastatic HER2+ breast cancer	Atezolizumab + trastuzumab + vinorelbine
NCT03595592 APTneo	Phase 3, open label	Active, not recruiting	650	Early high-risk and locally advanced HER2+ breast cancer	Arm 1:Trastuzumab + pertuzumab + carboplatin + paclitaxel (HPCT). Arm 2: Doxorubicin + cyclophosphamide (AC) followed by HPCT + atezolizumab, Arm 3: HPCT + atezolizumab
	**Durvalumab (anti PD-L1 antibody)**				
NCT02649686 CCTG IND.229	Phase 1, open label	Completed	15	Metastatic HER2+ breast cancer receiving trastuzumab	Durvalumab + trastuzumab [210]
NCT04538742 DB-07	Phase 1b/2, open label	Recruiting	450	Metastatic HER2+ breast cancer	Trastuzumab Deruxtecan (T-DXd) in Combination With Other Anti-cancer Agents
	**Avelumab (anti PD-L1 antibody)**				
NCT01772004 JAVELIN solid tumor	Phase 1, open label	Completed	1756	Metastatic or locally advaced solid tumors	Avelumab monotherapy to 26 HER2+ breast cancer [211]
NCT03414658 AVIATOR	Phase 2, open label	Recruiting	100	Advanced HER2+ breast cancer	Trastuzumab + vinorelbine with avelumab or avelumab + utomilumab (anti CD137)
	**Monalizumab (anti-NKG2A antibody)**				
NCT04307329 MIMOSA	Phase 2, open label	Active, not recruiting	38	Metastatic HER2+ breast cancer	Molalizumab + trastuzumab in cohort of low TILS (<5%) or cohort of high TILS (≥5%) [212]
**IMM2902 (HER2/SIRPα Bispecific mAb-Trap Antibody-receptor Fusion Protein)**					
NCT05076591	Phase 1, open label	Recruiting	135	Advanced solid tumors HER2+	IMM2902, dose escalation
**Utomilumab (anti-CD137 antibody)**					
NCT03414658 AVIATOR	Phase 2, open label	Recruiting	100	Advanced HER2+ breast cancer	Trastuzumab + vinorelbine with avelumab or avelumab + utomilumab (anti CD137)
NCT03364348	Phase 1, open label	Completed	18	Advanced HER2+ breast cancer	Utomilumab + T-DM1 or trastuzumab

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
