# Peer review of "Emerging Targeted Therapies for HER2-Positive Breast Cancer"

_cancers, 2023, doi:10.3390/cancers15071987_

Round 1

Reviewer 1 Report

The review article by Mercogliano et al provides a comprehensive overview of the current treatment in HER2 positive breast cancer. While it is easy to follow the overall structure of the article, and it is generally well written, there are some minor issues that should be addressed before being accepted for publication:

1.     Based on line 74, the focus of the article is on currently-tested HER2-tageted therapies. But, the section that immediately follows is on trastuzumab, the established first course of treatment for patients. This should be harmonized.

2.     HER2 does not need a ligand for homodimerization, which is unclear from the text in lines 45-47.

3.     Fig 1 can be larger (space permitting by the journal). The authors should also consider labeling the domain regions of HER2 receptor, and the respective antibodies close to their binding domain.

4.     The immunotherapy treatment is summarized in a table. For consistency, would it not be helpful to also summarize the other treatments types in a table?

5.     The authors specify when they are referring to HER2+ tumors that express the estrogen receptor (luminal subtypes). When not mentioned, does HER2+ subtype strictly refer to HER2+ tumors that are ER and/or PR negative?

6.     Please write that PRS-343 is a bispecific HER2/CD137 antibody in line 622 (where there are two periods, one of which should be deleted).

7.     There are some specific grammar and spelling errors, as well as formatting issues to be corrected/clarified:

a)                    It appears as if different sections were written by different people, which is expected for such a review, but grammar (childs rather than children - line 755’; ‘has proved’ should be ‘has proven’ – line 504), spelling (doble rather than double in table 1); paragraph structure and tone should be checked throughout the article for consistency and accuracy.

b)                    Structure of section 1.5 is difficult to follow. Its subsection title is HER2 and CD3 BsAb, but the section also includes a paragraph on CD16. Subheadings can contain more detail that fits with the content of the subsequent section.

c)                     p95HER2 in line 212 has a period before and after it. Is this a subheading for a subsequent paragraph? 

d)                    It is difficult to understand the structure with the current headings, in that it appears that some of the headings are meant to be subheadings. For example, HER2/HER2 BsAb appears that it should be a subheading of section 1.2 on Bispecific Antibodies, and should be labeled 1.2.1 rather than 1.3. Subsequently, section 1.3 should then address antibody-drug conjugates, section 1.4 TKI (which should be spelt out as done for ADCs in the previous section), section 1.5 Immunotherapy, with will then have the following subheadings: 1.5.1 on PD-1/PDL-1 antibodies, 1.5.2 on Immunotherapy enhancing ADCC and ADCP; 1.5.3 on Cell Therapies with CAR-T as a further subheading under (as the last sentence under cell therapies is In this section, we will address the 685 current cell therapies available for HER2+ breast cancer’).

e)                    The first paragraph of section 1.9 (line 499) is difficult to follow. In addition, a few introducing sentences to PD-1/PDL-1 would be helpful, as this section contrasts the sections on CAR-T/CAR-NK/CAR-M that are extremely detailed and information rich. The level of details provided in the different sections should feel balanced. 

f)  Should line 631 be a subheading? Also, please control the grammar from line 631 until the end of the section (line 670).

g)                    In a few instances, citation(s) is/are not given at the position where a study is first referenced, but sentences later where it is unclear if the same of another study is being cited.

Author Response

Answer to Reviewer #1

The review article by Mercogliano et al provides a comprehensive overview of the current treatment in HER2 positive breast cancer. While it is easy to follow the overall structure of the article, and it is generally well written, there are some minor issues that should be addressed before being accepted for publication:

We greatly appreciate the Reviewer’s comments and thorough read of the manuscript.

  1. Based on line 74, the focus of the article is on currently-tested HER2-tageted therapies. But, the section that immediately follows is on trastuzumab, the established first course of treatment for patients. This should be harmonized.

This point was resolved adding to the text that the work will not only describe the therapies currently being tested in the clinical setting but it will also describe the standard of care treatment.

  1. HER2 does not need a ligand for homodimerization, which is unclear from the text in lines 45-47.

The description of the receptor as an orphan receptor points to the lack of known ligands. For clarification this was stated in the revised manuscript.

  1. Fig 1 can be larger (space permitting by the journal). The authors should also consider labeling the domain regions of HER2 receptor, and the respective antibodies close to their binding domain.

The figure has been changed to explicit the domains on the HER2 molecule.

  1. The immunotherapy treatment is summarized in a table. For consistency, would it not be helpful to also summarize the other treatments types in a table?

In the revised manuscript new tables have been included summarizing the HER2-targeted therapies that are being tested in HER2+ breast cancer in the clinic. Table 1 consists of bispecific antibodies, Table 2 ADCs and Table 3 TKIs.

  1. The authors specify when they are referring to HER2+ tumors that express the estrogen receptor (luminal subtypes). When not mentioned, does HER2+ subtype strictly refer to HER2+ tumors that are ER and/or PR negative?

The term HER2+ is used broadly referring to HER2+ breast cancers with or without hormone receptor expression.

  1. Please write that PRS-343 is a bispecific HER2/CD137 antibody in line 622 (where there are two periods, one of which should be deleted).

This was corrected as PRS-343 was not defined in the text. Now it clearly states that PRS-343 is a bispecific antibody consisting of an anti-HER2 and an anti-CD137.

  1. There are some specific grammar and spelling errors, as well as formatting issues to be corrected/clarified:

  1. a) It appears as if different sections were written by different people, which is expected for such a review, but grammar (childs rather than children - line 755’; ‘has proved’ should be ‘has proven’ – line 504), spelling (doble rather than double in table 1); paragraph structure and tone should be checked throughout the article for consistency and accuracy.

The errors were corrected and the manuscript was proofread.

  1. b) Structure of section 1.5 is difficult to follow. Its subsection title is HER2 and CD3 BsAb, but the section also includes a paragraph on CD16. Subheadings can contain more detail that fits with the content of the subsequent section.

This has been corrected as the sections have been renumbered. In particular, this BsAb has its own section.

  1. c) p95HER2 in line 212 has a period before and after it. Is this a subheading for a subsequent paragraph?

It is a subsection and it has been corrected accordingly.

  1. d) It is difficult to understand the structure with the current headings, in that it appears that some of the headings are meant to be subheadings. For example, HER2/HER2 BsAb appears that it should be a subheading of section 1.2 on Bispecific Antibodies, and should be labeled 1.2.1 rather than 1.3. Subsequently, section 1.3 should then address antibody-drug conjugates, section 1.4 TKI (which should be spelt out as done for ADCs in the previous section), section 1.5 Immunotherapy, with will then have the following subheadings: 1.5.1 on PD-1/PDL-1 antibodies, 1.5.2 on Immunotherapy enhancing ADCC and ADCP; 1.5.3 on Cell Therapies with CAR-T as a further subheading under (as the last sentence under cell therapies is ‘In this section, we will address the 685 current cell therapies available for HER2+ breast cancer’).

The original manuscript had only the headings and subheadings in bold or italics, the numbering was added by the editorial office. It has now been corrected.

  1. e) The first paragraph of section 1.9 (line 499) is difficult to follow. In addition, a few introducing sentences to PD-1/PDL-1 would be helpful, as this section contrasts the sections on CAR-T/CAR-NK/CAR-M that are extremely detailed and information rich. The level of details provided in the different sections should feel balanced.

This section was completed as it was submitted in the revised version.

  1. f) Should line 631 be a subheading? Also, please control the grammar from line 631 until the end of the section (line 670).

The lines indicated do not contain a subheading.

  1. g) In a few instances, citation(s) is/are not given at the position where a study is first referenced, but sentences later where it is unclear if the same of another study is being cited.

The references have been checked and the position has been corrected.

Reviewer 2 Report

This review on anti-HER2 therapies in breast cancer is very interesting. It delves into the topicality of the subject with an extensive and current bibliography.
My only point is in the way the subsections are divided. The authors use only point 1, subdividing it into 1.1; 1.2; etc...
In my opinion, it would be more informative to subdivide the review into:
1. Introduction
2. Anti-HER2 therapies
  2.2 Antibodies
     2.2.1 Monoclonal Antibodies
     2.2.2 Bispecific Antibodies
....
2.3 TKI
2.4 Immunotherapy
...
X Conclusions and Future Directions

Author Response

Answer to Reviewer #2

This review on anti-HER2 therapies in breast cancer is very interesting. It delves into the topicality of the subject with an extensive and current bibliography.

My only point is in the way the subsections are divided. The authors use only point 1, subdividing it into 1.1; 1.2; etc...

In my opinion, it would be more informative to subdivide the review into:

  1. Introduction
  2. Anti-HER2 therapies

  2.2 Antibodies

     2.2.1 Monoclonal Antibodies

     2.2.2 Bispecific Antibodies

....

2.3 TKI

2.4 Immunotherapy

...

X Conclusions and Future Directions

We kindly appreciate the Reviewer’s evaluation. The changes to enumerate the sections were made in the revised manuscript.

Reviewer 3 Report

Dear author

Thank you for the submission of your article to our journal. I enjoyed your paper. Your paper discusses a wide range of subjects, including various monoclonal antibodies starting with trastuzumab, antibody-drug conjugates, TKIs, immunotherapy starting with PD-1/PD-L1 antibodies, and vaccine therapy. Your paper clearly shows the findings of each drug based on basic research and the results of the corresponding clinical trials. Doctors learning about anti-HER2 drugs for the first time can efficiently absorb knowledge, and specialists can organize their knowledge. Your paper is very timely that many anti-HER2 agents have appeared in clinical settings and clinicians are having trouble deciding how to use various anti-HER2 agents.Your paper should be very informative to our readers.

Author Response

Answer to Reviewer #3

Dear author

Thank you for the submission of your article to our journal. I enjoyed your paper. Your paper discusses a wide range of subjects, including various monoclonal antibodies starting with trastuzumab, antibody-drug conjugates, TKIs, immunotherapy starting with PD-1/PD-L1 antibodies, and vaccine therapy. Your paper clearly shows the findings of each drug based on basic research and the results of the corresponding clinical trials. Doctors learning about anti-HER2 drugs for the first time can efficiently absorb knowledge, and specialists can organize their knowledge. Your paper is very timely that many anti-HER2 agents have appeared in clinical settings and clinicians are having trouble deciding how to use various anti-HER2 agents.Your paper should be very informative to our readers.

We greatly appreciate the Reviewer’s comment.
